# River predisposition to ice jams: a simplified geospatial model

Stéphane De Munck[1], Yves Gauthier[1], Monique Bernier[1], Karem Chokmani[1], and Serge Légaré[2]

1 Centre Eau Terre Environnement, Institut National de la Recherche Scientifique (INRS), Quebec City (Quebec), G1K 9A9, Canada

2 Ministère de la Sécurité publique (MSP), Quebec City (Quebec), G1V 2L2, Canada

*Correspondence to:* Y. Gauthier (yves.gauthier@ete.inrs.ca)

**Abstract.** Floods resulting from river ice jams pose a great risk to many riverside municipalities in Canada. The location of an ice jam is mainly influenced by channel morphology. The goal of this work was therefore to develop a simplified geospatial model to estimate the predisposition of a river channel to ice jams. Rather than predicting river ice break up, the main question here was to predict where the broken ice is susceptible to jam based on the river's geomorphological characteristics. Thus, six parameters referred to potential causes for ice jams in the literature were initially selected: presence of an island, narrowing of the channel, high sinuosity, presence of a bridge, confluence of rivers, and slope break. A GIS-based tool was used to generate the aforementioned factors over regular-spaced segments along the entire channel using available geospatial data. An "Ice Jam Predisposition Index" (IJPI) was calculated by combining the weighted optimal factors. Three Canadian rivers (Province of Quebec) were chosen as test sites. The resulting maps were assessed from historical observations and local knowledge. Results show 77% of the observed ice jam sites on record occurred in river sections that the model considered as having high or medium predisposition. This leaves 23% of false negative errors (missed occurrence). Between 7% and 11% of the highly "predisposed" river sections did not have an ice jam on record (false-positive cases). Results, limitations and potential improvements are discussed.

## 1    Introduction

Ice jams result from the accumulation of fragmented ice on a section of a river, obstructing the channel and restricting the flow. Ice jams mainly occur during the breakup season but can also form in the period of freeze-up or even during winter when rain events cause a sudden increase of water levels and a dismantlement of the ice cover. The resulting floods can be socio-economically costly as well as life threatening (Beltaos and Prowse, 2001; Environment Canada, 2011). Many attempts have been made to develop reliable forecasting methods in order to provide early warnings and to mitigate the impacts of such events (White, 2003; Mahabir et al, 2007; White, 2009). However, existing forecast models are often site-specific: they combine numerous and complex triggering meteorological, hydrological and morphological factors (White,

2003; Beltaos, 2009; Bergeron et al, 2011). Moreover, when breakup occurs and ice starts to move downstream, another key question is: where would the released ice be susceptible to jamming? The goal of this study is to provide some answers to the aforementioned question by developing a simplified geospatial model that would estimate the predisposition of a river channel to ice jams. This is not a physical model simulating the processes of ice jamming but rather an approach based on some common knowledge about the general causes of ice jams and their relationship to the morphological characteristics of the channel, within a 2D spatial representation (De Munck et al., 2011). Being developed for an eventual application over large areas and multiple rivers, the geospatial model uses simplifications and provide a "first level" assessment of the predisposition to ice jam along the river channel. It has been developed on three Canadian rivers from the Province of Quebec: the Chaudière River, the Saint-François River, and the L'Assomption River (Figure 1), which all flow to the Saint-Lawrence River. They each have a history of ice jams and relatively frequent flooding of riverside municipalities. The Chaudière and Saint-François rivers flow mostly northward, through the geological areas of the Appalachians and of the Saint-Lawrence lowlands. Their length and drainage area are comparable: 185 km over 6 682 km² for the Chaudière River and 210 km over 10 230 km² for the Saint-François River. The L'Assomption River flows 200 km southward over the Canadian Shield and the Saint-Lawrence lowlands. It drains a 4220 km² watershed.

## 2    Background

According to Shen and Lianwu (2003), the key mechanism of the initiation of an ice jam at a river section is the convergence of ice motion, or when the incoming ice discharge exceeds the outgoing ice discharge. The convergence of ice flow can be produced by the reduction in driving forces and the increase in resistance forces to the ice motion when the ice run is not impeded by an intact ice cover. Both changes in driving and resistance forces are governed by the river geometry. In the literature, there is a consensus about the channel characteristics which can result in a reduction of ice transport capacity. Shen and Lianwu (2003) say that a reduction in channel slope or an increase in channel cross-section area, that is, a reduction in current velocity, will reduce the driving forces. On the other hand, a reduction in channel top width, the existence of meandering and braided sections, and shoals or islands in the channel will increase the resistance to the ice flow. According to US Army Corp of Engineers (2002), any river section where the slope decreases is a possible location for ice jamming. During freeze-up, the slower moving reaches freeze first and so will have a thicker ice cover come breakup. Another possible location might be a constriction in the channel, either natural, such as at a bend or at islands, or at man-made features, such as bridges. A third typical location is a shallow reach, where the ice can freeze to bottom bars or boulders and will not be lifted and moved by the increased water flow. According to Beltaos (1995; 2008), theoretical analysis and experience suggest that sharp bends, sudden reduction in slope, or constrictions, are frequent ice jamming sites, along with areas where the ice cover may be relatively thick and strong. According to Environment Canada (2011), there are

locations which are more susceptible to ice jam formation than others. These include the confluence of two rivers, channel constrictions, sharp bends, islands, bridge piers, shallow river reaches, the edge of a solid ice cover, and at sudden changes in the slope of the water surface. Often ice jams are caused by a combination of two or more of these factors. According to Ettema et al (1999), by virtue of their role in connecting channels and thereby concentrating ice within a watershed, confluences are perceived as locations especially prone to the occurrence of ice jams. According to Lindenschmidt and Das (2015), narrower, steeper and relatively straight channels are more susceptible to initiate breakup along the river. On the opposite, wider and mild slope sections of the river may have a persistent ice cover until the end of breakup. Therefore, the presence of an intact ice cover downstream would increase the risk of ice jams.

We can therefore summarize the key parameters leading to ice congestion and ice jam as:

- Reduction in channel slope or slope break
- Reduction in channel top width (naturally or due to border ice)
- Constriction in the channel from bends, meandering, islands, bridges
- Presence of shallow reaches and bottom bars
- Presence of an intact ice cover.

We should add that although ice congestion is the key parameter that leads to ice jams, an ice run can also simply be stopped by an obstacle, such as an intact ice cover, a bridge or an island.

Kalinin (2008) conducted a qualitative and quantitative study of several parameters mentioned above. On the rivers of the Votkinsk reservoir catchment (Russia), he found that a narrowing of the channel was present in 90% of the ice jams reported, islands were present in 80% of cases and bends were there in 70% as well. He also observed that the simultaneous presence of at least two of these five factors is characteristic of frequent ice jams.

To estimate a channel predisposition to ice jam, we will therefore consider: narrowing of the channel, sinuosity, presence of an island, presence of a bridge, confluence of rivers and slope breaks. These parameters are based on simple and relatively stable morphological characteristics and can be derived from easily available geospatial data. Shallow reaches and bottom bars are linked to water depth, which is variable throughout the year. The presence of an intact ice cover is also variable through time. For this reason, and because that bathymetry and ice maps are not available on a large scale, these two parameters will not be considered in this study. However, they are often linked to the morphological characteristics of the river channel (Turcotte and Morse, 2013).

## 3    Methods

### 3.1    Geospatialization of the selected parameters

In this work, "geospatialization" is the spatial representation of a physical characteristic of the channel and its transformation into a potential ice-jamming factor. This was done using a standard Geographical Information System (ArcGIS software) and some specific tools developed in ArcObject through the FRAZIL project (Gauthier and al., 2008) for the support of winter hydraulic modeling and ice-jam early warning systems. These tools enable the determination of the river channel centerline, its segmentation into equal length sections, calculation of the width, and calculation of channel sinuosity along the axis (Figure 2). Calculations were integrated along segments of equal length. Sections of 250m were found to be optimal considering the scale at which channel characteristics vary and the size of ice jams (hundreds of meters to kilometers (Beltaos, 2008)). Shorter sections overestimate narrowing and underestimate sinuosity. Long sections tend to underestimate narrowing and the impact of small features (islands, bridges). A variable length based on the homogeneity of the channel morphology could be an interesting avenue but it would have to be developed as a separate study.

Therefore, for the Chaudière, Saint-François and L'Assomption rivers, 444, 861, and 508 sections of 250m were created respectively. Input data came from CanVec, a digital cartographic reference product of Natural Resources Canada (Natural Resources Canada, 2016), and the Quebec Topographic Database (BDTQ) (Énergie et Ressources Naturelles Québec, 2008). The planimetric accuracy of these dataset is better than 2m. Shapefile layers include river channel, watershed, vegetated islands, bridges, rapids and elevations.

Note that for some rivers, when the upstream channel becomes very narrow, data representation can change from a polygon to a line and hence, we do not apply the model pass this point. Metadata do not indicate the minimal channel width represented by polygons in the dataset. But for the three rivers in this study we calculated that all sections in a polygon format were over 20m wide, which would be the limitation of the model if using this data source. Therefore, for the Chaudière River, the model was applied over the last 110 km to the St-Lawrence River. For the L'Assomption River, it was applied over the last 127km.

### 3.1.1    Narrowing Index (NI)

Considering that an ice jam formation is often due to a combination of different factors, our model proposes to combine and weight different parameters. Four parameters are first considered: 1) natural changes in the channel width, 2) presence of bridges, 3) presence of islands and 4) presence of an incoming tributary. They are linked to ice jamming processes for distinct physical reasons. However, for simplification, we consider in the model that they are all contributing to congestion through narrowing of the channel section available for ice transit.

For example, islands generally involve a narrowing of the main channel (Banshchikova, 2008) as well as a breaking slope
from steep to mild. Thus moving ice is forced to slow down and to obstruct the channel. The model would therefore consider
this section as predisposed to ice jamming. The drawback of this generalization is that the model assumes that an island
located in the middle of the channel has the same impact on restricting the ice movement than an island closer to the shore.
We did try to consider the specific location, type, size and shape of the islands but the complexity of dealing with these
combined parameters was generating more uncertainties in the model results. We should also mention that with this
approach, the model does not take into account the potential release of some pressure when ice is pushed into secondary
channels.

The need for simplification also applied to bridges. A bridge is an obstacle which disturbs the natural flow of ice moving
downstream, specifically when pillars are close to each other. According to Urroz et al. (1994), the ratio of the distance
between pillars by the channel width has to be high in order to have a smaller impact on the moving ice process. The
interaction between ice and bridges is a balance between ice-driving and ice-resisting forces (Beltaos, 2006). Bridges can act
as an obstacle or a constraint. From a hydraulic point of view, the pillars of a bridge divide the main channel into several
narrow channels, where the ice is more susceptible to jam. Again, considering the presence of a bridge as a narrowing of the
channel enables the model to infer some predisposition to ice jamming on this specific section. And specifying a certain
width reduction permits to adjust the impact of the bridges. Here, we consider that a half reduction of the channel width
when a bridge is crossing the river would give a substantial weight to bridges in the final predisposition model. The available
datasets in this study do not specify the characteristics of the bridges (type of bridge, number and shape of pillars).
Therefore, the drawback of this generalization is that all bridges are considered equal. However, a user could adjust the
width reduction parameter to better fit a specific river. And bridges which characteristics do not pose a risk of ice jamming
could simply be removed from the input layer.

The final parameter that has to be generalized is the tributary. Small rivers usually respond more quickly to rising run off
compared to large rivers. A quick hydrological response in tributaries may trigger an early breakup and send an ice run into
the main channel. Since the ice cover of the main channel is likely to still be intact, the ice run can stop at the confluence,
become an immediate ice jam or initiate an ice jam during the breakup to come on the main channel. Literature considers
that the major impact of a tributary is the potential input of ice (or even sediment) into the main channel that would also
result in reducing the available space or would create an obstacle for ice transport in that main channel. Again,
conceptualizing the tributary as a narrowing of the main channel allows the model to infer a predisposition for ice jamming
on this section while the specified width reduction determines the importance of the impact. Here, the width reduction is
equal to the minimal width of the tributary at the outlet (Figure 3). This gives more importance to large tributaries.

Even if we fit many parameters into a unique narrowing index, each parameter is calculated independently and its relative
importance can be adjusted. In the end, the Narrowing Index is calculated from the natural or adjusted channel width of each
250m section. When the width of the preceding section is smaller than that of the actual section (sections 5 and 6 in Figure 4)
the index has a value of 1 (no narrowing). When the width of the preceding section is larger than that of the actual section
(sections 2 to 4 in Figure 4), the narrowing index is obtained by dividing the width of the actual section by the closer
upstream maximum width. A value tending towards 0 will indicate a stronger narrowing of the channel. It should be noted
that although a narrowing of the channel can in some instances concentrate energy and favor transit of ice runs, the model
only considers it as an aggravating factor.

### 3.1.2    Sinuosity Index (SI)

Bends and loops are known to increase resistance to the ice flow (Shen and Lianwu, 2003). Due to preferential flow, ice is
deported towards the concave bank and may start accumulating there, gradually reaching the opposite bank and creating a
jam (Zufelt, 1988). Here it should be noted that the simplified model do not consider the fact that the first bend of a
meandering reach is more likely to initiate an ice jam.

The Frazil toolbox (Gauthier and al., 2008) is used to obtain a standardized sinuosity coefficient. It uses the *Sinuosity4*
equation proposed by Dutton (1999) to express the sinuosity coefficient (SV) in values ranging between 0 and 1 (Eq. (1)).

$$\text{Sinuosity4} = \sqrt{1 - \frac{1}{SV^2}} \qquad\qquad\qquad (1)$$

where, SV is the curvilinear distance between two points divided by the direct linear distance between the same two points.
Calculations of SV are based on inflection points, which separate two curves going in opposite directions. A 0-value for
*Sinuosity4* means that there is no sinuosity in the section. The distance between two inflexion points can cover adjacent
250m sections. The calculated sinuosity is applied to all sections it overlays. If a section was overlaid by two different values
of sinuosity, the mean value was calculated and retained.

### 3.1.3    Slope Break Index

A change of the river bed slope from steep to mild is the typical case involved in ice jams. Since gravity is the driving force,
the ice can lose its energy when it reaches a milder slope, and can stall or arch across the river and initiate an ice jam
(Wuebben and Gagnon, 1995). Such a change of slope is also present at the estuary of a river or at lakes and reservoirs,
where ice jams often form (Saint-Laurent et al., 2001). On a technical point of view, this parameter should be easy to
integrate to the model. A slope break index would be calculated based on the approximate channel surface altimetry from a
Digital Elevation Model (DEM) (Eq. (2)).

$$\text{Slope Break Index} = \Delta \frac{\Delta\,Height}{\Delta\,Length} \tag{2}$$

Initially, we did consider this parameter in the model. The data from the 1:20 000 Quebec Topographical database are built over contour lines with a ±5 meter resolution. This coarse resolution resulted in shaky slope break index values, giving an

175 inadequate representation of the actual river slope. Complete bathymetric data for the rivers under study were not available. For this reason, this version of the model did not integrate the slope break index. However, an accurate LIDAR model could be used if available. If a future version of the model integrates the slope parameter, it should also include rapids, since ice jams almost never initiate in rapids but often, at the end. It is nonetheless possible to force a low predisposition to sections with rapids.

### 3.1.4 Ice jam dataset

The ice jam dataset is provided by the Quebec Ministry of Public Safety (MSP). The data comes from digital or paper event reports provided by local authorities under the jurisdiction of MSP (Données Québec, 2016). The database contains ice jams reported in the province of Quebec from 1985 to 2016, with approximate geolocation since most jams are longer than a single coordinate and because this geolocation does not refer to the toe where the jamming process is initiated. The database

is not "validated" in the sense that each event has not been compared to corresponding hydrographs, that a few observed ice jams could be related to anchor ice or frazil, and that reported locations do not necessarily refer to the toe where the jamming process is initiated. Therefore, validation of the model from this database is not absolute. But it is nonetheless a unique source of information in Canada. Although proceeding to a complete validation of the database was out of the scope of this study, we have discarded observations that could not be located with enough accuracy. Furthermore, the analysis will

consider not only the sections directly coinciding with an ice jam observation, but also neighbouring sections where the toe could have formed. In this study, we focused on 118 historical observations: 61 ice jam reports for the Chaudière River, 33 for the St-François River and 24 for the L'Assomption River. The 61 ice jams listed on the Chaudière River were used as test sites for calibration of the conceptual model. Then validation of the model was performed over the other two rivers.

### 3.2 Conceptual model on the Chaudière River

The conceptual model proposed here integrates the narrowing index and the sinuosity index to establish the potential predisposition of a river channel to ice jams, the "Ice Jam Predisposition Index", IJPI.

### 3.2.1 Standardization of the index values

First, we standardized the range of values for each index. Each index was transposed into four classes, 0 to 3, from the weakest to the strongest impact on ice jam predisposition. The thresholds between these classes were determined using a K-Means clustering approach. The model was developed mainly with the data from the Chaudière River. However, to determine the thresholds for the classes of Narrowing and Sinuosity index using K-means, we used the entire range of values from the three rivers in this study in order to provide a more robust and representative model. Four clusters were created with squared Euclidian distances, replicated 5 times. Table 1 shows the thresholds established from the K-Means approach.

### 3.2.2 Weighting of the index values

The narrowing and sinuosity indices may contribute differently to the ice jamming process. To determine the weight of each index in the conceptual model, we have used the same approach as Kalinin (2008), which is to cross reference the ice jam occurrence from the historical dataset with the values of both indices at these sites. The ice jam occurrences on the Chaudière River were categorized into three classes: the "frequent" category was assigned to a section where at least two ice jams were listed in the dataset, while the "occasional" category was assigned to sections where only one ice jam was listed. Sections with no ice jam recorded were classified in the "rare" category. We then compared the frequent and occasional occurrences with the values of the narrowing and sinuosity indices at these river sections. As shown in Table 2, the Narrowing Index usually outnumbers the Sinuosity Index, indicating that it should have a more important weight in the model. If we cross reference sections with a frequent occurrence of ice jams with sections where both indices show the maximum value (class #3), we would obtain a ratio of 1.5 in favor of the Narrowing index. If we cross reference all sections where an ice jam was observed, with sections where both indices show a moderate or high value (class #2 and class #3), we also obtain a ratio of 1.5 in favor of the Narrowing Index. A multi-criteria analysis (Saaty, 1990) then assigns a weight of 0.60 to the Narrowing Index, and a weight of 0.40 to the Sinuosity Index.

### 3.2.3 Ice Jam Predisposition Index (IJPI)

The final step of the model is the calculation of the Ice Jam Predisposition Index (IJPI). The standardized class value (V) attributed to each index (k) is multiplied by the weight factor (W) for that index. The sum of weighted values is divided by the sum of weighted maximal values (Eq. (3))

$$\text{Ice Jam Predisposition Index (IJPI)} = \frac{\sum_{k=1}^{2} V_k W_k}{\sum_{k=1}^{2} V_{max} W_k} \tag{3}$$

According to the maximum value (Vmax = 3) and the normalized weight factor, equation 3 can be simplified to Eq. (4).

Ice Jam Predisposition Index (IJPI) $= \frac{\sum_{k=1}^{2} V_k W_k}{3}$ (4)

The values resulting from the ice jam predisposition index (IJPI) range from 0 (no predisposition to ice jam) to 1 (very high predisposition to ice jam). Table 3 shows the 14 possible IJPI values obtained from equation 4. We used boxplots to study the statistical distribution of the IJPI values, on sections of the Chaudière River with listed ice jams and on sections without ice jam (Figure 5). To simplify the results of the model into three main classes (high, medium and low predisposition to ice jams), we used the median and the third quartiles of IJPI values as thresholds: IJPI≥0.54; 0.40≤IJPI<0.54; IJPI<0.40.

## 4 Results and discussion

### 4.1 Chaudière River

Figure 6 shows the results of the model applied on all 250 m sections of the Chaudière River (calibration site). High predisposition is shown in red, medium predisposition in orange and low predisposition in green. Locations of reported ice jams are indicated with thumbtacks. The symbol is blue (correct assessment) when the ice jam falls into a section with a
235 medium or high predisposition. It is magenta (false-negative error) when the reported ice jam falls into a section with a low predisposition. Again, we have to keep in mind that there may be a difference between the initiation site (higher predisposition) and the observation site (anywhere along the jam). In contrast, a false-positive error would give a high value of predisposition in a section where no ice jam was observed. This doesn't mean that the model is necessarily wrong. It is possible that ice jams on some of these sections have never been reported. It is often the case in isolated or non-vulnerable
areas. Or, since the model gives a "predisposition", it doesn't mean that an ice jam will automatically occur or as already occurred. So the false-positive results are to be considered objectively.

In total (Table 4), the model indicates that 51 of the 444 sections (11%) would have a high predisposition for ice jams, 69 sections (16%) would have a medium predisposition and 324 sections would be at low risk. Of the 61 reported ice jams on the Chaudière River, 20 (33%) are located on sections with a high predisposition, 23 (38%) are on a section with a moderate
predisposition and 18 (29%) are on sections with low predisposition. These 18 sightings represent the false-negative results or where the predisposition to ice jamming was underestimated.

Table 4 also shows that of these 18 cases, 3 are related to the presence of a major tributary in the section. This could indicate that the value applied for channel width reduction underestimates the actual impact of a tributary on ice jamming.

However, we also have to look at these results in the context of the uncertainty related to the geolocation and length of the
250 ice jam reported in the historical database. Considering that an ice jam may have a length of a few hundred meters to a few kilometers, one could have reported the sighting upstream from the toe of the jam, where it was initiated. Therefore, the geolocation of the point in the database may lie upstream of the predisposed section. It is interesting to see that for 10 of the

18 false-negative errors on the Chaudière River, the ice jam is reported less than 1km upstream from a section with a high or medium predisposition. So we may even underestimate the performance of the simplified model, although it is impossible to confirm without more accurate data.

Table 4 finally shows that over the river, 32 sections (7%) were classified with having a high predisposition to ice jams but without any event reported. For moderate predisposition, it concerns 46 sections (10%). These cases are the false-positive results. As mentioned earlier, these are not necessarily errors. But it is also probable that the model is overestimating predisposition in some areas. For example, when looking at the false-positives cases (32 sections of high predisposition), we can determine that in each case the class of the Narrowing Index is greater than the class of the Sinuosity Index. This would indicate that we may overestimate the impact of the narrowing of the channel. The false-positives can be caused by all types of narrowing but we found that 5 of the "faulty" sections have a bridge. Considering that there are only a dozen of bridges in the study area, this number tends to confirm that all bridges are not equal and that the model could be easily improved at the local level with specific information about the bridge characteristics.

## 4.2    Saint-François River

As mentioned earlier, the Saint-François River is comparable to the Chaudière River. Both flow mostly northward, through the same geological region, and have similar channel length and drainage areas. Results for the IJPI on the St-François River are shown in Figure 7. As can be seen in Table 5, the percentage of sections respectively classified as high, moderate or low predisposition to ice jams are similar to the Chaudière River. Of the 33 reported ice jams on the St-François River, 11 (33%) are located on sections with a high predisposition, 13 (40%) are on a section with a moderate predisposition and 9 (27%) are on sections with low predisposition (false-negatives). Here we notice that three false-negative errors occurred on sections with at least one island.

The number of sections with false-positive results is similar to the Chaudière River also (17% vs 20%). However, when looking at the false-positives errors with a high predisposition index (79 sections), only 46 show that the Narrowing Index is greater than the class of the Sinuosity Index. But again, the number of "faulty" sections with the presence of a bridge (12) is quite high since there are around 20 bridges on the St-François River.

In Figure 8, we take a closer look at some of the false-negative errors on the St-François River. These omissions are more significant in terms of public safety. They could be caused by a parameter not considered in the model (e.g. slope break), by the simplification approach, or by an inaccurate geolocation of the observation. In site A (Figure 8) the shape of the channel and the presence of islands are probably enough to trigger an ice jam. But the islands are small and do not seem to impact on the mean narrowing calculated over the sections. The problem is then related to generalization and scale. In A', we can see that the area has shallow waters (not considered in the model), which could also support ice jamming. In site B, the model

sees the bends upstream and downstream but again, misses the islands. Here they are located in a wider section of the channel, cancelling the narrowing effect. In site C, the first island causes a sudden narrowing well detected by the model. But the main channel width remains stable over the next sections, again cancelling the potential narrowing impact of the islands. However, the changes in direction of the main channel should have increase predisposition. In site D, the land strip going into the channel could have caused the ice jam. But the feature is so localized compared to the section's length that it may not sufficiently affect the mean width to register as a narrowing. Thus, the error here could be related to scale. Finally, let's note that if we can't identify with certainty the cause of a false-negative error, either from channel characteristics, model generalization or scale, there is still the possibility that the ice jam was initiated by the presence of an intact ice cover. According to Lindenschmidt and Das (2015), wider and mild slope sections of the river are more susceptible to have a persistent ice cover until the end of breakup. Again, this is a time dependent parameter, which is not considered by the model. Again, there is also the possibility that the observation point did not correspond to the ice jam toe.

### 4.3    L'Assomption River

We could expect to get different results from the L'Assomption River, as it flows southward in a different geological area and has a much higher sinuosity in its lower portion. Figure 9 shows the results of IJPI on the L'Assomption River. According to Table 6 the percentage of sections classified as having a high or moderate predisposition to ice jams is higher by about 20% compared to the two other rivers. Of the 24 reported ice jams on the L'Assomption River, 14 (58%) are located on sections with a high predisposition and 10 (42%) are on a section with a moderate predisposition. There are no false-negatives. However, with more sections at risk (meandering channel) and a smaller ice jam dataset, false-positive errors are naturally higher (31%). As expected, more false-positive errors are on sections where the Sinuosity Index is greater than the Narrowing Index. Finally, as for the other rivers, the bridges seem to create some overestimation of the risk as 8 false-positives are at sections where a bridge is present (on a total of 14 bridges over the study area).

### 5    Conclusions

A geospatial model for estimating a river's predisposition to ice jams was proposed based on the key morphologic parameters leading to ice jams. For simplification of the model, four factors were integrated into a single Narrowing index: natural narrowing, presence of islands and bridges and incoming tributaries. A Sinuosity index was also calculated. Each index was standardized and given a weight. Calibration was done on the Chaudière River and validation was performed on the Saint-François and L'Assomption Rivers in Quebec, Canada. The model was setup using 250m long river sections. The development and validation phases were supported by the ice jam database of the Quebec Ministry of Public Safety, with historical observations from 1985 to 2016. This database presents a certain degree of uncertainty, particularly concerning

the location of some of the reported ice jams (toe and length). It is nonetheless a great tool to document areas at risk of ice jams and to assess the reliability of the proposed model.

The model produced between 11 and 19% of river sections classed as having a high predisposition to ice jams and between 15 and 26% of river sections presenting a moderate risk. When compared to the historical observations, most reported ice jams fall into these sections (71% on the Chaudière River, 73% on the St-François River and 100% on the L'Assomption River). Ice jams that occurred on low predisposition areas are called false-negative errors. The uncertain geolocation of the reported ice jams may account for part of these. A majority of false-negative errors are located less than 1 kilometer from sections with a higher predisposition. However, results tend to show that the model underestimates the role of islands and tributaries in the initiation of an ice jam. Some errors left unexplained could be related to time dependent parameters not integrated into the model such as water depth or the presence of an intact ice cover, although both are indirectly related to channel morphology.

River sections that are categorized with a predisposition to ice jam but where no ice jam was reported are called false-positive cases (17% for the Chaudière River, 20% for the St-François River and 31% for the L'Assomption River). They are not necessarily errors since the historical dataset is not exhaustive and because a predisposition is not a certainty. However, the results show that the model could overestimate the impact of bridges.

Overall, the results of this geospatial model are very promising. Even in a conceptual form and by using only parameters that are mostly stable over time, the model seems to correctly represent the nature of the river and the areas where the morphology has an impact on ice jam occurrence. Having applied the model over three different rivers also ensures a certain degree of transferability to the approach.

However, it is important to understand some limitations of the model. First, it is not developed for freeze-up ice jams occurring from frazil accumulation or hanging dams. It addresses ice jams following a breakup event. Also, it is a simplified model, intended to work with data easily available for most rivers. It does not simulate the physical processes of ice jams but rather locate areas where the morphology of the channel presents some characteristics known to initiate ice jams. The model gives a first level assessment of the ice jam potential of rivers. Some fine tuning could have to be done if high resolution data or local knowledge is available on a specific river, in order to better take into account some local and more complex causes of ice jams.

Even in its present version, the model is already providing valuable information to the Quebec Ministry of Public Safety and to the municipalities located along the studied rivers. In addition to forecasting potential ice jam flooding sites, an improved version of the model could bring information for land planning, zoning, bridge construction or insurance evaluation. In the province of Quebec, the historical database is a great tool to document areas at risk of ice jams. The geospatial model is now

a complementary tool to map these areas, as well as others for which no ice jam has yet been reported. And the model is a valuable tool for provinces or countries where no ice jam database exists.

For a future version of the model, potential developments could be:

- To consider attenuating factors, such as a section located immediately downstream a reservoir or directly within a rapid;
- To consider the width, shape and length of the contributing reach upstream from a predisposed section (is there potentially enough incoming ice to produce a jam?);
- To consider sudden channel widening (dissipation of the energy and ice run stalling);
- To take into account the presence of hydraulic structures (weirs, dams, dam reservoirs, etc.);
- To test the model using the US Ice Jam database (Carr et al, 2015)
- And certainly to use the slope index, upon availability of accurate elevation data.

The authors are presently starting the application of the model on all rivers prone to ice jams in the province of Quebec. They are also planning the work on a new version that will be integrated within an ice jam vigilance and alert system, combining spatial predisposition, temporal forecasting and ice status.

**Competing interests**

The authors declare that they have no conflict of interest.

**Acknowledgements**

The research presented here was funded by a NSERC Discovery Grant (2009-2014) to Dr. Bernier (INRS). The setup of the ice jam dataset was funded by the Quebec Ministry of Public Safety (MSP). The authors would also like to thank Jimmy Poulin and Fatou Sene from INRS and Nicolas Gignac from the Quebec Ministry of Public Safety for their contribution.

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

**Table 1: Threshold values for the Narrowing (NI) and Sinuosity (SI) indices, as determined by the K-Means approach.**

|  | Narrowing Index (NI) | Sinuosity Index (SI) |
| --- | --- | --- |
| Class 0 |  |  |
|  | 0.56 | 0.24 |
| Class 1 |  |  |
|  | 0.77 | 0.46 |
| Class 2 |  |  |
|  | 0.92 | 0.69 |
| Class 3 |  |  |

**Table 2: Comparison of river sections with reported ice jam events (frequent and occasional occurrences) with the narrowing and sinuosity indices at these locations. NI/SI is the ratio of the Narrowing Index (NI) on the Sinuosity Index (SI).**

| Ice jams | Number of sections with a high Narrowing Index (NI) | | Number of sections with a high Sinuosity Index (SI) | | Ratio NI/SI | |
|---|---|---|---|---|---|---|
| | Class 2 | Class 3 | Class 2 | Class 3 | Class 2 | Class 3 |
| Frequent | 5 | 3 | 3 | 2 | 1,67 | 1,5 |
| Occasional | 4 | 4 | 3 | 5 | 1,33 | 0,8 |

**Table 3: Possible results from the Ice Jam Predisposition Index (IJPI). Values highlighted in light grey (IJPI≥0.40) were selected as representing a moderate predisposition to ice jams while values highlighted in dark grey (IJPI≥0.54) would represent a strong predisposition to ice jams.**

| Sinuosity Index / Narrowing Index | 0 | 1 | 2 | 3 |
|---|---|---|---|---|
| 0 | 0 | 0.20 | 0.40 | 0.60 |
| 1 | 0.13 | 0.33 | 0.53 | 0.73 |
| 2 | 0.26 | 0.46 | 0.66 | 0.86 |
| 3 | 0.40 | 0.60 | 0.80 | 1 |

 **Table 4: Results and accuracy of the IJPI on the Chaudière River**

| Model results | Number of river sections | | Reported ice jams |
|---|---|---|---|
| High predisposition | 51/444 | (11%) | 20 (33%) |
| Medium predisposition | 69/444 | (16%) | 23 (38%) |
| Low predisposition | 324/444 | (73%) | 18 (29%) |

**False negative errors**

Features present on river sections with false-negative errors

| Bridge | Island | Tributary | No specific feature |
|---|---|---|---|
| - | - | 3 | 15 |

**False-positive errors**

| River sections with high predisposition but no ice jam reported | 32/444 | 7% |
|---|---|---|
| River sections with medium predisposition but no ice jam reported | 46/444 | 10% |

**Table 5: Results and accuracy of the IJPI on the Saint-François River**

| Model results | Number of river sections | | Reported ice jams |
|---|---|---|---|
| High predisposition | 93/861 | 11% | 11 (33%) |
| Medium predisposition | 132/861 | 15% | 13 (40%) |
| Low predisposition | 636/861 | 74% | **9 (27%)** |
| **False negative errors** | | | |
| Parameters present on river sections with false-negative errors | | | |

| Bridge | Island | Tributary | No specific feature | |
|---|---|---|---|---|
| **-** | 3 | - | 6 | |

| **False-positive errors** | | |
|---|---|---|
| River sections with high predisposition but no ice jam reported | 79/861 | 9% |
| River sections with medium predisposition but no ice jam reported | 96/861 | 11% |

**Table 6: Results and accuracy of the IJPI on the L'Assomption River**

| Model results | Number of river sections | | Reported ice jams |
|---|---|---|---|
| High predisposition | 98/508 | 19% | 14 (58%) |
| Medium predisposition | 133/508 | 26% | 10 (42%) |
| Low predisposition | 277/508 | 55% | 0 |
| **False-positive errors** | | | |
| River sections with high predisposition but no ice jam reported | 55/508 | 11% | |
| River sections with medium predisposition but no ice jam reported | 100/508 | 20% | |

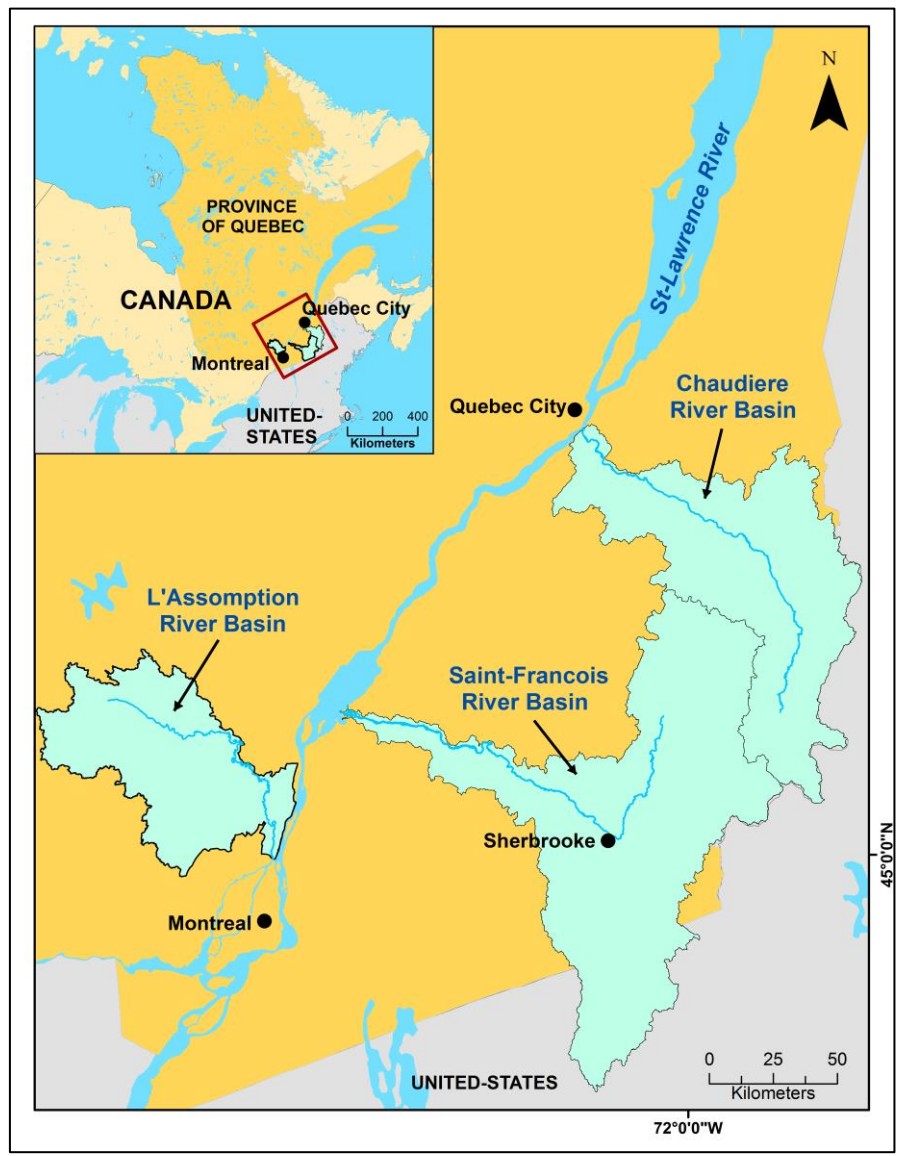

**Figure 1: Location of the Chaudière River, Saint-François River and L'Assomption River (Province of Quebec, Canada).**

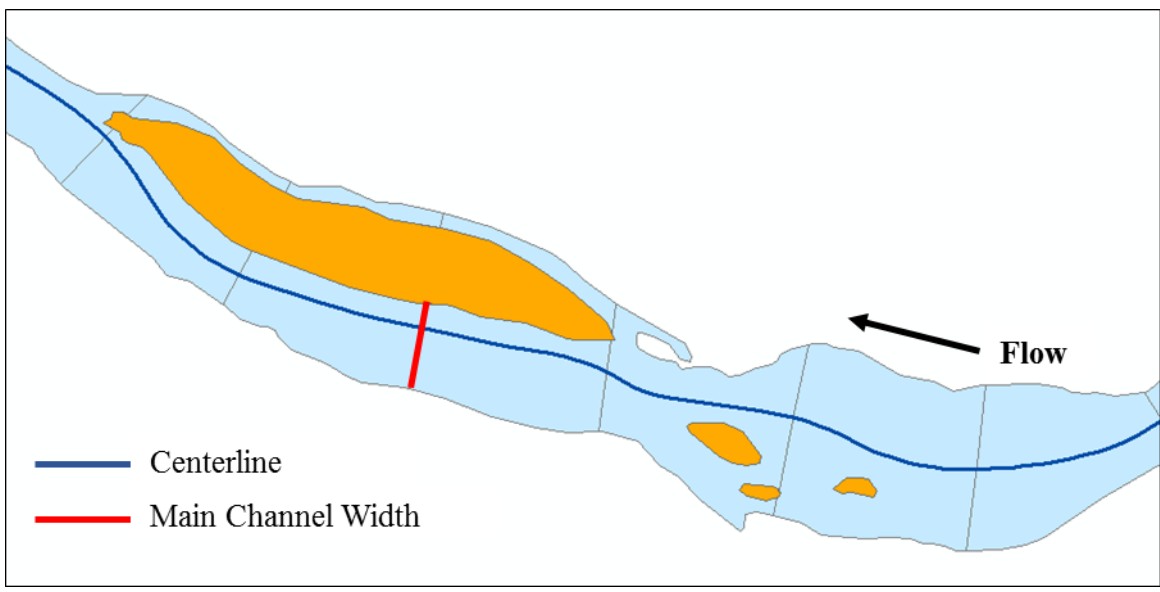

**Figure 2: Spatial representation of the channel centerline, channel width and channel 250m sections in presence of islands (from the FRAZIL tools).**

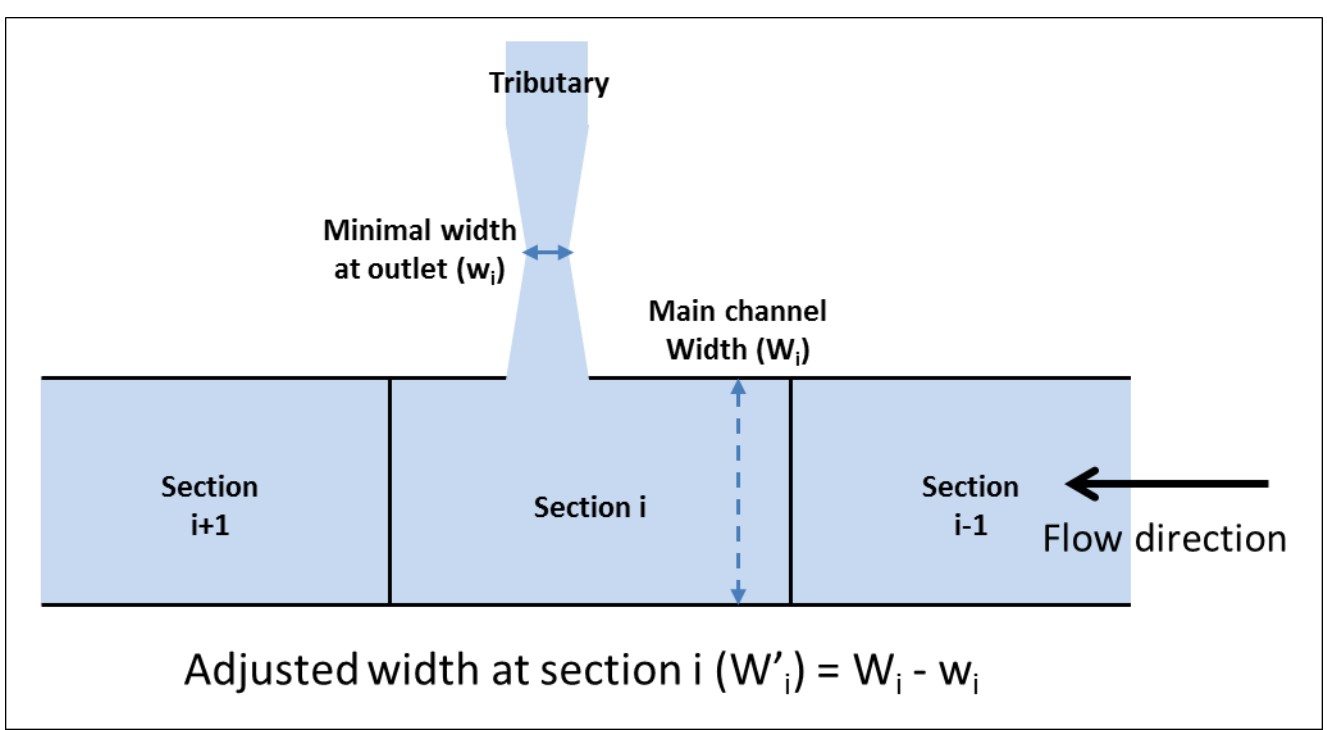

**Figure 3: Modification of the channel's width due to incoming tributaries. For segment i, the adjusted width ($W'_i$) is obtained by subtracting the minimal width of the tributary at the outlet ($w_i$), from the main channel width ($W_i$).**

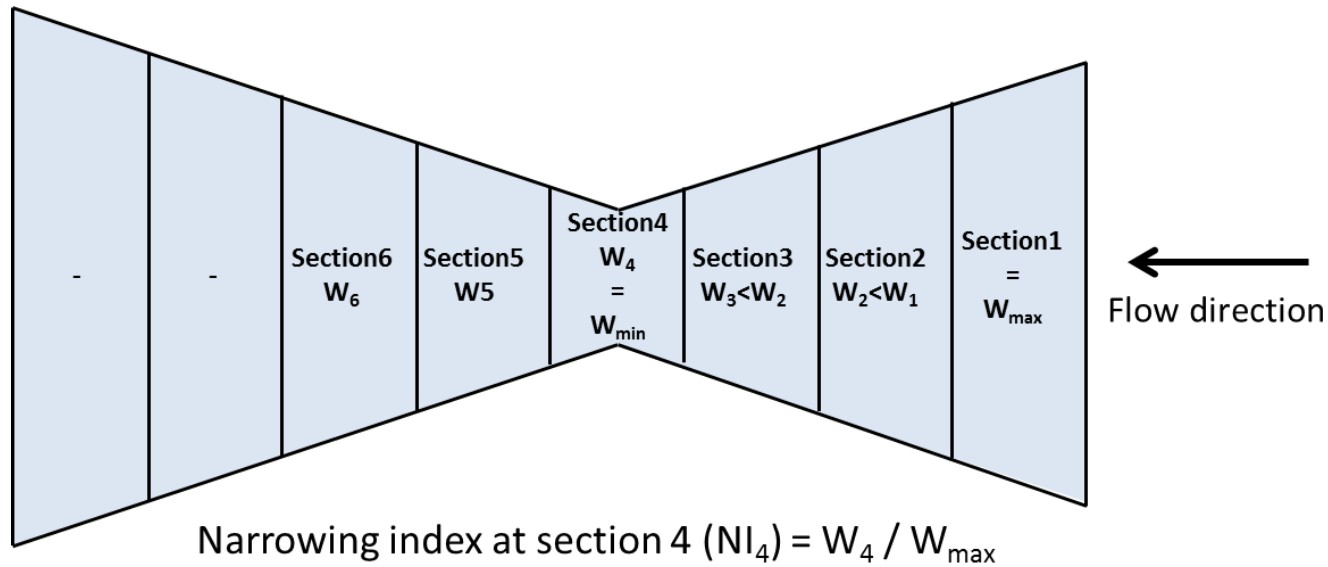

Figure 4: Approach used to calculate the Narrowing index, dividing the section's width (W) by the upstream maximum width (Wmax).

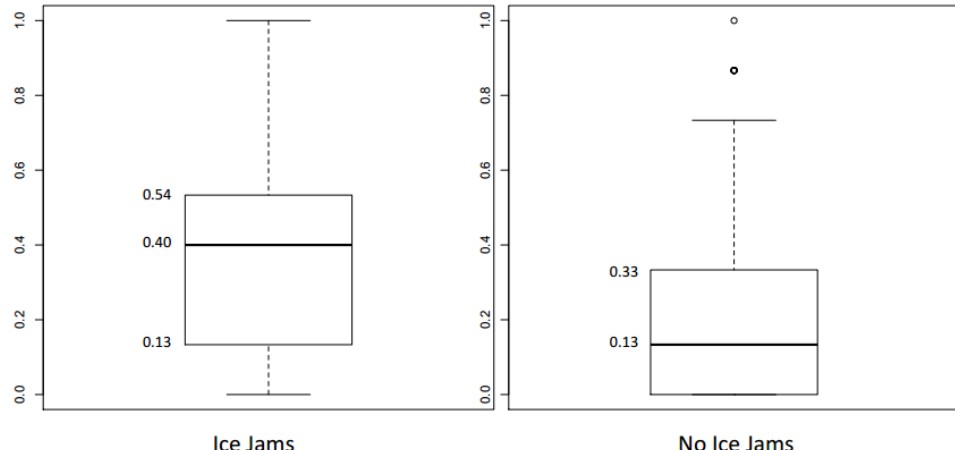

**Figure 5: Boxplot of the IJPI values on the Chaudière River. Graph on the left is for 250m river sections where ice jams were reported. Graph on the right is for river sections with no ice jam listed. Numbers represent the median, first and third quartiles.**

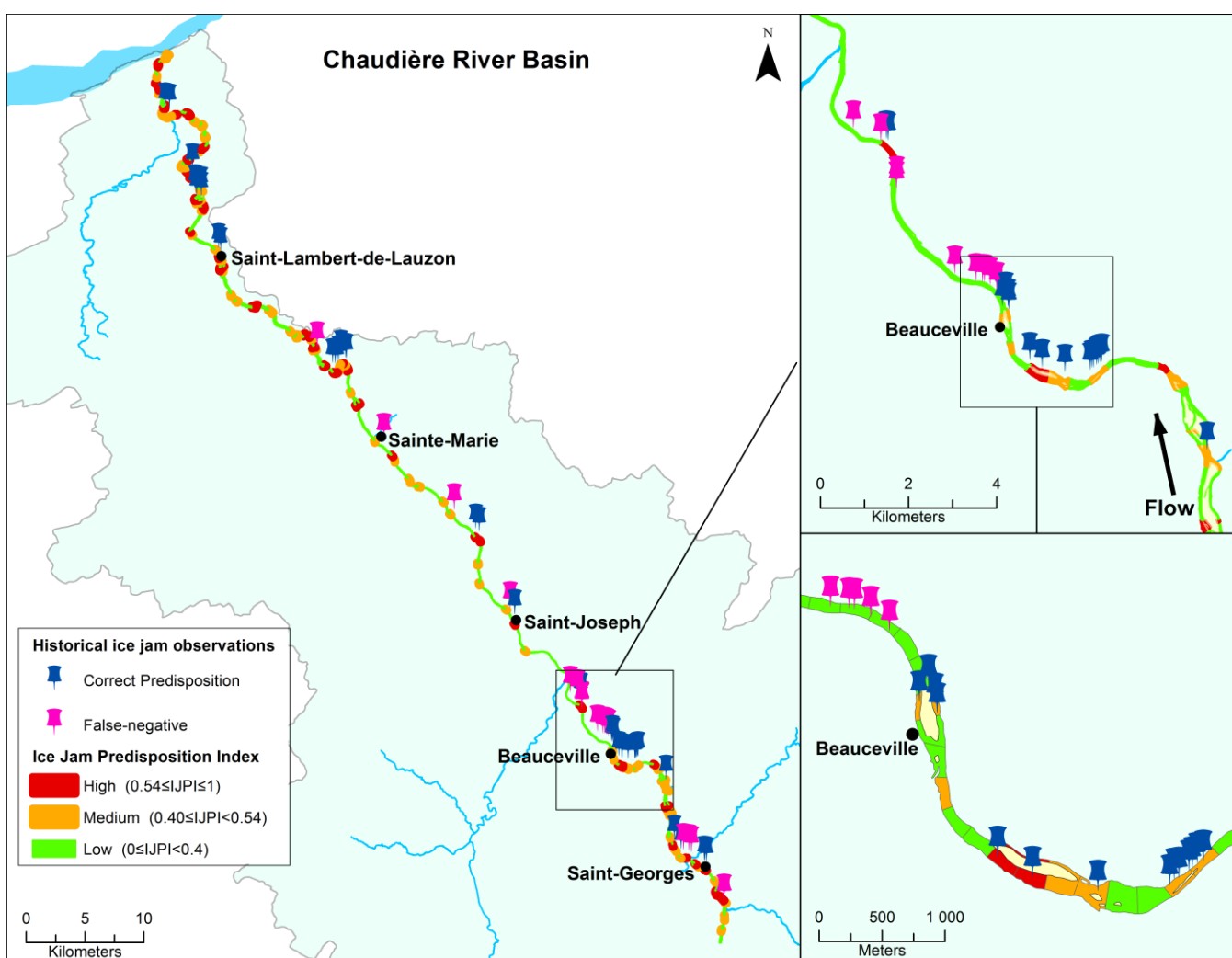

**Figure 6: Map of the model results on the Chaudière River, over 250m sections. Thumbnails are the locations of reported ice jams. Blue is used when the ice jam falls on a section with a moderate to high predisposition (correct assessment). Magenta is used when the ice jam falls on a section with a low predisposition (false-negative error).**

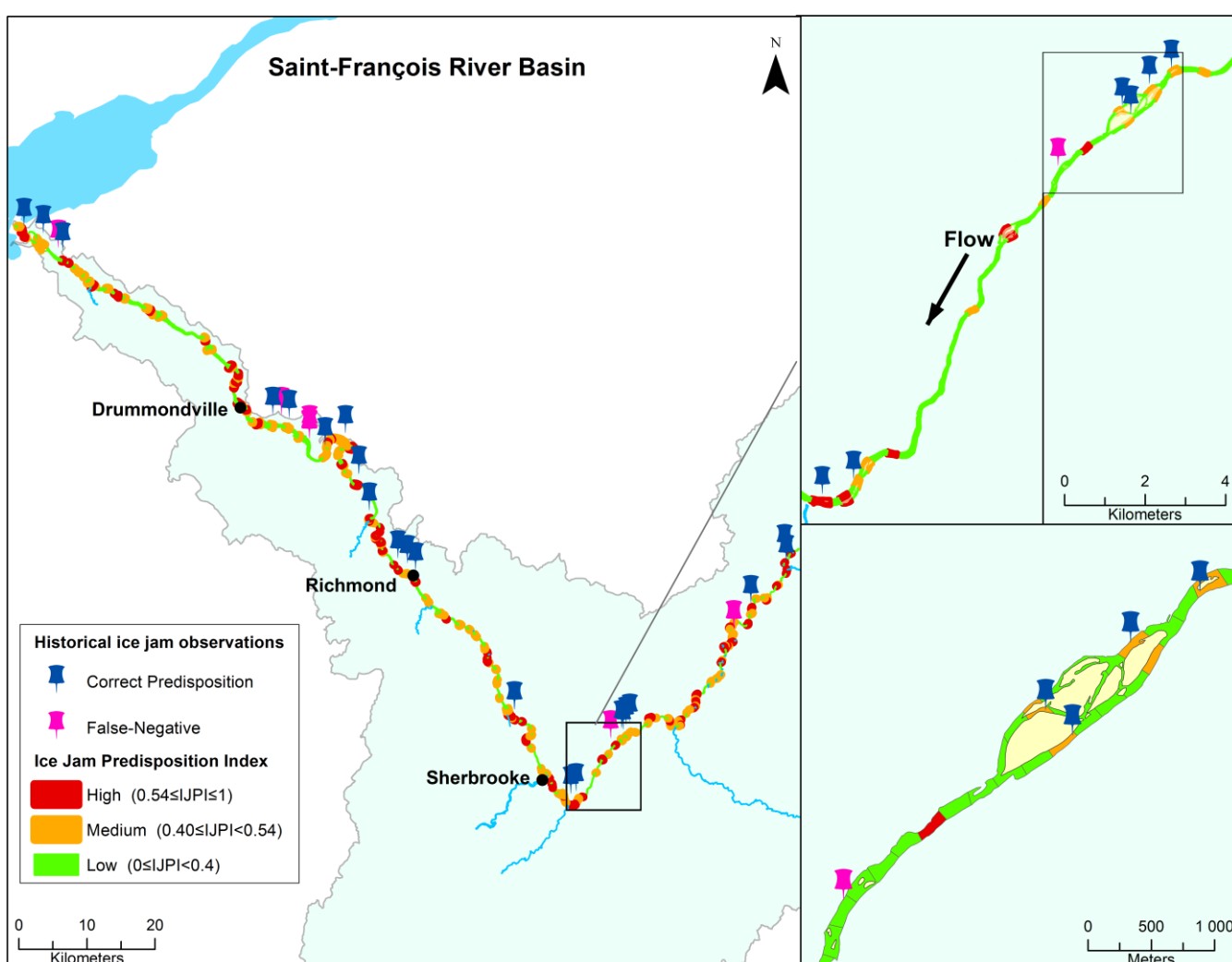

**Figure 7: Map of the model results on the St-François River, over 250m sections. Thumbnails are the locations of reported ice jams. Blue is used when the ice jam falls on a section with a moderate to high predisposition (correct assessment). Magenta is used when the ice jam falls on a section with a low predisposition (false-negative error).**

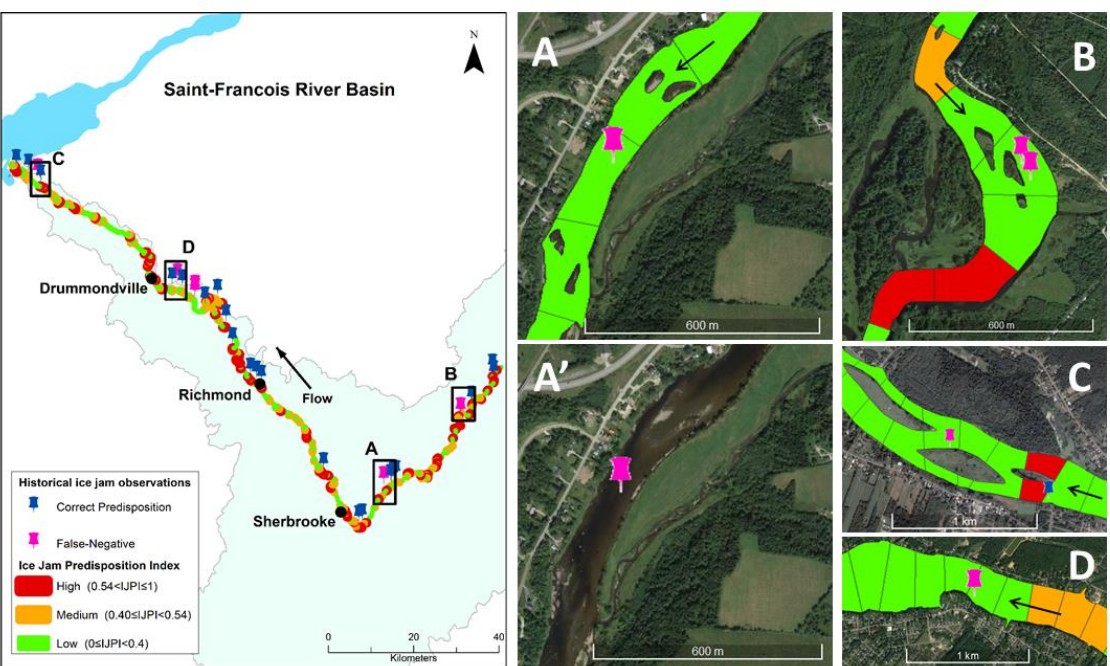

**Figure 8: Examples of false-negative errors on the St-François River for site A, B, C and D.**

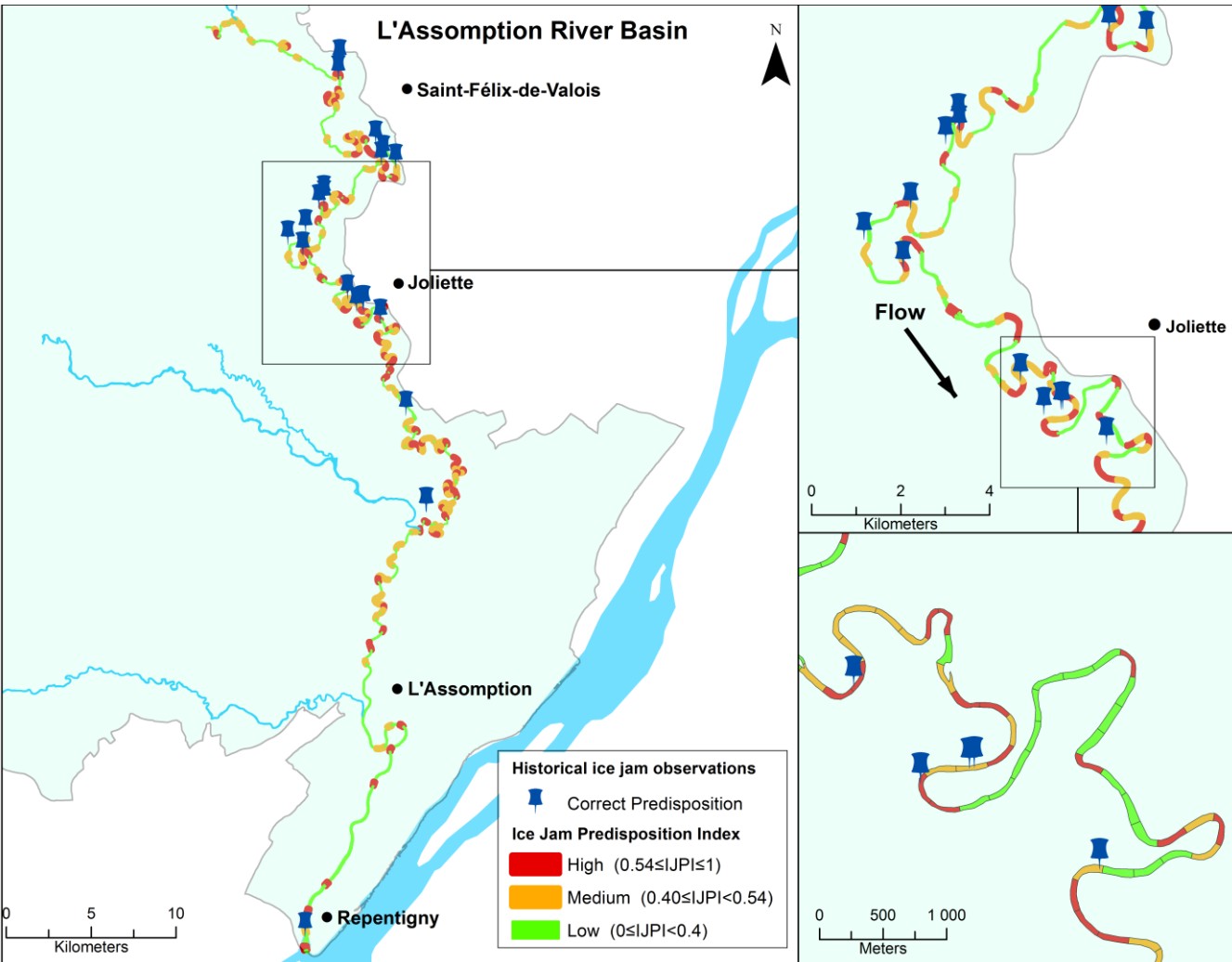

**Figure 9: Map of the model results on the L'Assomption River, over 250m sections. Thumbnails are the locations of reported ice jams. Blue is used when the ice jam falls on a section with a moderate to high predisposition (correct assessment). Magenta is used when the ice jam falls on a section with a low predisposition (false-negative error).**