# Peer review of "River predisposition to ice jams: a simplified geospatial model"

_Natural Hazards and Earth System Sciences, 2016_

## Referee Comment (RC1) · Anonymous Referee #1 · 14 Jan 2017

Review of Manuscript 2016-308 "River predisposition to ice jams: a simplified geospatial model"

General comments This paper explores ice jam predispositions along northern rivers using a geospatial modelling approach in which sets of fluvial geomorphological parameters are compared with ice jam occurrences. There is a high success rate of predicting ice jam locations, however some errors do occur due to the presence of sand bars and low water depths, variables not considered in the model The approach does give a first assessment of the ice jam potential of rivers, hence, the paper is deemed publishable if the following minor revisions are considered.

Specific comments The narrowing index (NI) for bridge peers is rather arbitrarily derived that can lead to over- or under-estimation of their effect on ice jamming. No considera-
tion was given to the number of peers spanning across the bridge. Hence, the NI of a suspended bridge would have the same NI value as a bridge with many closely spaced peers. Could you please give an explanation of why this wasn't considered?

Also on the subject of bridges, I find that bridge peers do not necessarily stop an ice run to create an ice jam but reduce the inertia of the ice run enough for it to slow down and stop at a location further downstream from a bridge peer. Would such a consideration improve the predictability of the model?

Technical corrections Line 11: change "has been" to "was' Line 13: change "have been" to "were" Line 21: change "precipitation" to "rain events" Line 26: change "jam" to "jamming" Line 66: change "jam" to "jamming" Line 71: change "opposite" to "on the other hand" Line 89: change "comes" to "came" Line 92: change "channel's representation goes at some point" to "channel representation changes" Line 94: include "was represented by polygons and" after "channel" Line 113; the line should read "overestimates or underestimates ice jam occurrences at bridges." Line 114: "sediments" is not plural. Line 115: replace "transiting" to "transport" Lines 115-116: The clause should read "To approximate this parameter in the model, . . ." Line 166: "compared" (past tense) Line 172: The method is called "multi-criteria analysis" Line 202: replace "On the opposite" to "In contrast" Line 250: some errors here in the cross-referencing Line 259: replace "They mean that . . . something" with "This means that there is something not considered in the model". Line 259: change "seem" to "seems" Line 260: replace "grow bigger" to "extend further" Line 274: replace "it presents" with "has" Line 280: replace the last word "on" to "at" Line 305: replace "applied on" to "setup using" Line 311: replace "happened" to "occurred" Line 314: should "water depth" be included to bathymetry and the presence of an intact ice cover? Line 312: replace "exportability to" to "transferability of" Line 324: replace "false and positive errors" to "false-negative and false-positive errors".

[Figure]

2016.

---

## Referee Comment (RC2) · B. Turcotte (Referee) · 23 Jan 2017

Paper title: River predisposition to ice jams: a simplified geospatial model

Authors: DeMunck, Gauthier, Bernier, Chokmani, Légaré

General comments:

Note that another review has been submitted before mine, but I have not read it before performing my review. BT

This paper presents a very interesting and original model that uses qualitative, geospatial information to quantitatively identify channel locations where ice jams could form. Its development and calibration is supported by an ice jam database. Overall, it seems that there is a great potential for the development of a model that could be used to

identify potential ice jamming sites and this could be combined with a river ice breakup forecasting model.

However, the model overlooks or simplify a number of key ice jamming parameters, factors, processes, and information that may limit its reliability and the paper really presents what appears to be the early development stage of an acceptable model. It seems that the authors globally lack in experience and confidence (a lot of sentences seem defensive) and the following points (below) should be considered to improve the next versions of the model (the actual knowledge about river ice processes has only be partially considered). I consider that the actual version of the model is almost dangerous to use by public security services or for flood insurance purposes.

At this point, I am not sure if I recommend (1) that the authors should make multiple technical changes to the paper and include a discussion that mentions the many limitations of the model in its actual form or (2) that the authors should present a new paper with a more advanced version of the model that would address and include most of the following points. I would tend to vote 1 because I consider that the model is original and represent a step forward in the field of river ice and flood forecasting. In this case, I would encourage the authors to present an improved version of the paper with a serious discussion and to present, in the years to come, an improved version of the model that would potentially include a completely new model structure.

1. The authors do not mention that they have observed ice jams and do not refer to any experience in the field (e.g., to verify the sites presented at Figures 8 – 10 or to look under bridges if pillars can be pointed out for ice jamming) . Therefore, this research is only based on theory and the authors cannot really confirm that the model is reliable. Beyond the government ice jam data base, the authors should have conducted a complete historical research and confirmed that the mentioned ice jam dates corresponded to specific hydro-meteorological events. This type of data base often confuses ice jams with other ice processes that generate winter or spring flooding (e.g., anchor ice and hanging dams). Moreover, at locations where observation is not
easy, where there is no societal vulnerability, or where the jamming and release occur at night year after year, ice jams may have gone unnoticed (as somewhat mentioned in the paper).

2. The authors refer to particular factors influencing ice jamming, but do not seem to understand all the physics that link these factors with ice jam processes. The authors never refer to the distinction between the toe and head of an ice jam and they barely mention something about their potential length (that can be much greater than 250 m and therefore extend in sections that have nothing to do with the initiation of the jam). It is crucial to point out that the parameters influencing ice jamming sites refer to the toe (initiation site), which can be hundreds of meters or even kilometers away from the ice jam observation site. This can influence the results of the research positively or negatively.

3. A number of parameters such as channel widening (dissipation of the energy and ice run stalling), the presence of hydraulic structures (weirs, dams, dam reservoirs, etc.), and the presence of a tight, single bend (not a meander) have not been mentioned in the study and could help reducing false-negative errors. On the other end, it seems that channel narrowing is assumed to generate ice jamming but in some cases, the concentration of energy actually favors the transit of an ice run. From my point of view, trying to fit many parameters in a "narrowing equivalent" will limit the potential development of the model.

4. Obstacles and gradient variations could explain a significant ratio of ice jams. This may require a more sophisticated spatial analysis that may become tedious to automatize.

5. One important parameter affecting ice jamming is the potential quantity of ice, i.e., the contributing reach. If there is not enough ice to produce an ice jam that can affect the floodplain, the jam may remain unnoticed. The most critical jamming sites are located downstream of long sections where an ice run would simply not stop. This has

to be mentioned here and potentially included in a future version of the model.

6. The model could consider factors that prevent the formation of an ice jam (e.g., immediately downstream of a reservoir) and the model could gain in accuracy and reliability.

7. In the end, at this development stage of the model, for Quebec, the data base itself could represent a more reliable tool to identify potential ice jamming sites than the model calibrated with the data base.

Specific comments:

Abstract:

There is no introducing context in the abstract.

Line 7: "any" should be moderated. The model has been tested on three rivers only and a number of parameters and factors are not considered.

Line 8: Remove "up"

Lines 11 and 13: Should be "was" and "were"

Lines 16-17: I am not sure that talking about "false positives" is pertinent here. These are not really errors.

Introduction:

Line 19: "emerge" may not be appropriate

Line 19: why using "specific"

Line 21: "precipitation" should be "rain" or "runoff" could simply be used

Line 21: Why "partial"? Mid-winter events can be quite "complete" from my point of view, depending on the rain and breakup intensity

Line 22: I would say "can be" instead of "are" [. . .socio-economically costly]. Not all ice

jams have economic consequences. Additional references could be mentioned.

Line 25: More references could be added. That of Bergeron et al. is a possible one, among others.

Line 26: Rephrase

Line 31: I would say "relatively frequent flooding"

Figure 1: Second part of the Figure may not be necessary. There is a lot of empty space in the map north of Quebec City that could be used to increase the size of the legend or to rearrange the ratio of each sub-Figure. Please confirm that the L'Assomption River watershed is the right one. It seems that it is in contact with the St. Lawrence along 80 km... The southern part of the St-Francois River could be indicated approximately.

Background:

This section should be reorganized: The authors should mention that an ice jam can form because of congestion or because of an obstacle. The use of a transport capacity in the literature only represents one simplified interpretation that has been overused here. Most of this section refers to congestion processes as if an ice jam could only be the result of an unimpeded ice run (Jasek 2003) that slows down and stop. Indeed, an important portion of ice runs encounter a physical obstacle (such as an intact ice cover mentioned at the end) and suddenly stop. This has nothing to do with congestion or a "reduction in the ice transport capacity".

Line 39: I am not sure if Beltaos would refer to "volume of ice". I believe that it would be the "ice discharge".

Line 41: "were" should be "are"

Line 41: Authors should seek additional references. The books "River ice Jams" of "River ice Breakup" are potential sources of complementary information.

Lines 43-44: The authors should mention that there are different types of islands. They can be naked bars, vegetated bars or emerging rock outcrops (vegetated or not) and not all of them are associated with a break in the channel gradient. From my point of view, the presence of an island is associated with ice jam in part because the flow can bypass the congested channel and release the pressure on the impeded ice run, therefore leaving an ice jam on one side of the island. The authors only mention "narrowing" as the basic process to identify potential ice jamming sites.

Lines 45-47: "close" to each other. Not all bridges present pillars and pillars are often profiled to minimize to effect on flow conditions and ice transport capacity. Also, bridges are often build at natural (or artificial) narrows that already represent a limitation for ice mobilisation. This may affect the result of the study. Also, the flow often accelerates under a bridge because of the smaller river width and ice runs could easily transit that these locations.

Lines 48-50: Not well explained. The basic process may be that the flow along the concave bank drowns while the ice floats. Also, the authors mentioned that sinuosity may "initiate" a jam. This is correct, but it would mean that the first bend of a meandering reach is more likely to cause jamming that the last one. The authors could include this (or mention that this has been or should be considered) in their study.

Line 51-56: I would replace "narrow" and "wide" by "small" and "large". Remove "from precipitation". I would say something like "A quick hydrological response in tributaries may trigger an early breakup and send an ice run into the main channel". You refer to the milder gradient of the larger channel, but it is not very important if the ice cover is intact in the main channel. Also, note that the gradient is not part of this study. Two ice runs that meet at a confluence at the same time is quite unlikely. Is this what the authors refer with "merging ice runs"?

Line 58: "impetuous" could be "power" or "energy".

Lines 57-59: This is one of the most important parameters explaining ice jamming.

Lines 60-62: There are many types or gravel bars in gravel bed rivers. The authors only mention two types here (point bars or side bars). I am not sure that I agree with the reasoning presented: gravel bars are usually mobilized when the flow increases and stabilize when the flow decreases. How could they form and migrate if there was never a potential for transport? The first part of these lines does not need a reference as most people know about bars. It is really the second part of the sentence that needs the support of a reference.

Lines 63-64: This refers to the slope and should be presented with lines 57-59, together with reservoirs and lakes.

Lines 65-66: The authors should refer to the concept of an impeded ice run here (Jasek ,2003, or Jasek and Beltaos, 2008). This is the only parameter that does not directly refer to the morphology, but it is very important. This is why I would reorganize this entire section.

Line 68: Please mention what river and what is the dominant morphology?

Line 69-70: This is important because the authors use this single idea of the combination of two ice jamming factors for their model. I understand the need to simplify reality, but I am not sure that one publication can justify this choice.

Lines 70-74: I would refer to Bergeron et al. here. Note that this is already partially mentioned earlier.

Line 74: "However, in the present study"...

Line 74: I would remove "(mostly stable over time)" because you mention this two line below

Line 74: "will be" should be "are"

Line 77-78: I would express this differently. The word "dynamic" in the river ice literature usually refers to processes such as ice runs, ice jams and ice dams. Note that the depth

can be linked to the morphology and the cover characteristics as well (e.g., Turcotte and Morse, 2013). If the authors believe that the presence of bars is important, well their emergence is completely linked to the depth and discharge!

Methods:

Line 87: How does the 250 m in length compares with the channel width? Why not using a variable length that depends on the width of the channel or the homogeneity of the morphology and alignment? I guess that this would be complex for automatic interpretation to be performed and it would not fit with the title of the paper.

Line 88: What can be said here about the size of ice jams if the data base does not include such information?

Line 89-91: Is this precise enough to document parameters such as narrowing?

Line 92: Islands: Does the model differentiates bars and stable islands?

Line 92: rapids: This is very important and has not been mentioned before. Ice jams almost never initiate in rapids but often at the end of rapids. Does this includes riffles or just rapids?

Line 93: This is not very reassuring: The authors should mention that a width less than X m could not be included in the model for spatial information accuracy limitations. Then, it means that the model is actually not adaptable to small rivers.

Lines 100-102: I understand that the model is simplified for practical reasons. However, as noted in the general comments and as the authors mention at the end, these four factors are linked to ice jamming processes for distinct physical reasons.

Line 104: "will focus" should be "focuses"

Line 104-105: About the secondary channel presenting a more competent ice cover: I do not agree and the authors do not refer to any study to support this. From my point of view, there could be less (or no) ice in the secondary channel and at some location,

the secondary channel plays a determinant role in the ice jamming initiation process.

Line 105: "The model also assumes"

Line 105-107: Every island site is different and I am not sure that the simplification proposed by the authors is the most adequate one. Food for thought.

Line 108: Do you have any information about the pillars? What is the assumption here? If the bridge is located downstream of rapids, ice blocs may be small and easily pass under. If large is slabs come in contact with wide, rectangular pillars, yes, they might be stopped right there. This would be a serious engineering error that as no real link with a channel narrowing.

Line 109: What do you mean by "initially"? I believe that this factor should have been calibrated more accurately or considered differently (not a narrowing).

Line 112-113: I believe that this is a major mistake made by the authors: Presenting an assumption in the methodology, mentioning that it could be improved before the results are presented and not doing anything later despite this could have been better calibrated.

Line 116: Specify that this gives more importance to large tributaries. Again, this has almost nothing to do with channel narrowing.

Line 117: "In the end,"

Line 118: "the index has" not, "the index will have"

Line 120: Replace "by the most recent maximum width of the upstream sections" by "by the closer upstream maximum width"

Figure 3: Does not explain well how the tributary is considered

Figure 4: The flow direction should be the same than in Figure 3. A narrowing index (equation) should be presented for each presented section.

Line 131: "ranging from"

Line 131-132: This definition refers to SV, not the Sinuosity4... The authors should mention that SV is always larger than 1.

Line 133: I am not sure that "several" will satisfy the reader. Can you present a range? Can this take into account single bends and not only meanders?

Line 141: "did not integrate"

Line 139-142: Did the authors try to use the 5 m resolution? I am sure that some governmental agencies have data concerning river profiles and hydraulic models. This would probably not be precise enough to determine changes between 250 m sections, but it could very well identify slope breaks that are so important in jamming processes. As a reader, I am disappointed about this ending and this introduces a difficulty acceptable omission in the model. "Luckily" for the authors, a change in slope is normally characterized by a change in morphology and pattern and therefore slope breaks are somewhat indirectly covered by the model. Including gradient data would probably improve the model's result.

Line 145: Not pertinent to mention the 950 jams since it covers all the rivers and since it increases every year.

Line 146: I would say "approximate" since most jams are longer than a single coordinate and because this geolocation does not refer to the toe where the jamming process is initiated. In some instances, the toe could be kilometers away from the observation point. Also, as mentioned in the general comments, some reported ice jams could be intense anchor ice or frazil jam events and these processes could take place at locations where ice jams are not likely to form. A validation with a corresponding rising Q should be performed.

Line 157: This is the Chaudière River section but this sentence refers to the three rivers. Table 1: Put NI and SI in the title as well

Lines 163-166: Note here that you use "reported" ice jams to calibrate a model. Taking into account previous comments may improve the calibration result and their reliability.

Line 170: Remove "would"?

Line 172: Remove "would"?

Table 3: This is a fair analysis tool but note that this version of the model cannot gain a high precision potential in part because it is limited by a 2D IJPI. Also, note that highly sinuous reaches often present a relatively constant width and therefore, the two parameters considered here are not independent. Therefore, a value of 1 may be difficult to obtain in reality.

Results:

Line 201: About false negative errors: Please consider the potential length of the jam and the difference between the initiation site (predisposition) and the observation site (anywhere along the jam).

Line 202: About false positive errors: Not really an error because this is not an ice jam temporal prevision model. Please consider at everywhere in the paper that important ice jams can happen where there is no observation point nor vulnerability.

Line 208: Please replace the terminology "risk" by an appropriate word. This is not a synonym of predisposition.

Line 209: I understand that the model can underestimate some factors, but this is your calibration river. This could have been better considered if you were not limited to two indicators and if observations had been made in the field.

Lines 210-211: Exactly, and this should be stated clearly in the previous sections. Not only a source of incoming ice, but also a source of incoming runoff and javes.

Lines 212-222: Please eventually consider: The contributing area for ice blocks, the gradient, an increasing width, the absence of observation points along the river, the

absence of vulnerability along the river, and finally, ice scars on trees. You should present the potential reasons for false-positive errors in the form of bullets.

Line 217: "is also overestimating [the predisposition] in some areas".

Line 219: Same comment as line 209. Bridges are often located at natural narrows, but there design does not necessarily impede the transit of ice runs.

Line 224: How short? Please specify a range.

Table 4: You could have further investigated the 6 "no specific feature". This could mean factors that have not been considered.

Line 245: Note about bridges: Their presence can mean less snow ice and more thermal ice. Their presence can also be associated with de-icing salt falling on the ice surface. Then, what would be the final relative ice resistance at the time of breakup?

Figure 7: Please adjust the errors ("source de renvoi introuvable")

Table 5: Same comment as Table 5

Lines 259-263: The model may be missing something because there is no info about the gradient. A widening is also a site for ice jams, especially downstream of rapids or riffles. Note also that the analysis assumes that the toe of the jam was correctly located, which may not be true.

Line 260: Please had a reference that supports that sand bars are associated with ice jams. I do not know any.

Line 265-266: Yes. But then, this could be associated with other morphological or areal patterns.

Line 267: Please reconsider the use of "dynamic" parameter everywhere in the paper.

Line 274: The L'Assomption River is only sinuous in its lower portion.

Lines 278-280: A lot of sections may be associated with ice jamming, but the contributing area is just too small. The ice jam will most often form in the first (upstream) predisposition area located downstream of a long stretch of relatively fragile ice.

Figure 8: Potential interpretation (I do not know this site): This is enough narrowing, widening and changing direction to generate an ice jam. The low floodplain on the left bank and the possible secondary channel all support ice jamming. The marker may point a pool between two riffles where ice jams often form (against a small hanging dam...).

Figure 9: Potential interpretation: B: The ice run loses energy in the bend and loses further energy in the widening. C: Water evacuation channels and changes in direction, enough to initiate an ice jam.

Figure 10: Potential interpretation: This is an energy concentration area followed by a dissipation area. If there is no info about the jamming scenario (jamming of an impeded or unimpeded ice run), the reason for the ice jam event is hard to certify.

Figures 8 to 10: Note how all these reported jam are located where roads or houses are close to the river, ideal observation points that may not correspond to the ice jam toe.

Table 6: You could lower the False-Positives by considering the length of the potential contributing area.

Conclusions:

Line 299: "static" could be "morphologic"

Line 300 and Lines 302-303: Yes, but the slope could not be considered and this should be mentioned as a potential development, not as a limitation of the actual model. This should be part of a discussion section. Overall, some sentences could be written more positively and confidently.

Line 306-307: This sentence should be removed as it is already mentioned two lines

above.

Line 312: Define "very close" as opposed to "short distance"...

Line 314: Here, the bathymetry is considered as a dynamic parameter. Why? Why not just considering morphological (pattern, width, gradient, topography) and ice (type, potential thickness, possible processes) parameters?

Line 320: "interesting" should be "promising".

Lines 321-322: Do you refer to freezeup jams or to hanging dams?

Line 325: "more different" should be "additional"

Line 325: Please had a reference (Carr et al., 2015?) for the US database

Lines 327-328: Consider this change: "In addition to forecasting potential ice jam flooding sites, an improved version of the model could bring information for..."

Line 330: Consider this change: "combining spatial predisposition and temporal forecasting".

---

## Author Response (AR1)

[revised manuscript text omitted]

**Commentaire [GY12]:** Comment 2

**Commentaire [GY13]:** Comment 3

**Commentaire [GY14]:** Comment 3

**Commentaire [GY15]:** Comment 3

**Commentaire [GY16]:** Comment 9

**Commentaire [GY17]:** Comment 3
(+Entire section rewritten)

For example, islands generally involve a narrowing of the main channel (Banshchikova, 2008) as well as a breaking slope from steep to mild. Thus moving ice is forced to slow down and to obstruct the channel. The model would therefore consider this section as predisposed to ice jamming. The drawback of this generalization is that the model assumes that an island located in the middle of the channel has the same impact on restricting the ice movement than an island closer to the shore. We did try to consider the specific location, type, size and shape of the islands but the complexity of dealing with these combined parameters was generating more uncertainties in the model results. We should also mention that with this approach, the model does not take into account the potential release of some pressure when ice is pushed into secondary channels.

The need for simplification also applied to bridges. A bridge is an obstacle which disturbs the natural flow of ice moving downstream, specifically when pillars are close to each other. According to Urroz et al. (1994), the ratio of the distance between pillars by the channel width has to be high in order to have a smaller impact on the moving ice process. The interaction between ice and bridges is a balance between ice-driving and ice-resisting forces (Beltaos, 2006). Bridges can act as an obstacle or a constraint. From a hydraulic point of view, the pillars of a bridge divide the main channel into several narrow channels, where the ice is more susceptible to jam. Again, considering the presence of a bridge as a narrowing of the channel enables the model to infer some predisposition to ice jamming on this specific section. And specifying a certain width reduction permits to adjust the impact of the bridges. Here, we consider that a half reduction of the channel width when a bridge is crossing the river would give a substantial weight to bridges in the final predisposition model. The available datasets in this study do not specify the characteristics of the bridges (type of bridge, number and shape of pillars). Therefore, the drawback of this generalization is that all bridges are considered equal. However, a user could adjust the width reduction parameter to better fit a specific river. And bridges which characteristics do not pose a risk of ice jamming could simply be removed from the input layer.

The final parameter that has to be generalized is the tributary. Small rivers usually respond more quickly to rising run off compared to large rivers. A quick hydrological response in tributaries may trigger an early breakup and send an ice run into the main channel. Since the ice cover of the main channel is likely to still be intact, the ice run can stop at the confluence, become an immediate ice jam or initiate an ice jam during the breakup to come on the main channel. Literature considers that the major impact of a tributary is the potential input of ice (or even sediment) into the main channel that would also result in reducing the available space or would create an obstacle for ice transport in that main channel. Again, conceptualizing the tributary as a narrowing of the main channel allows the model to infer a predisposition for ice jamming on this section while the specified width reduction determines the importance of the impact. Here, the width reduction is equal to the minimal width of the tributary at the outlet (Figure 3). This gives more importance to large tributaries.

**Commentaire [GY18]:** Comments 37 (+Entire section rewritten)

**Commentaire [GY19]:** Comment 3 (+Entire section rewritten)

**Commentaire [GY20]:** Comments 38 (+Entire section rewritten)

And comment #2 and #3 from reviewer

**Commentaire [GY21]:** Comment 3 (+Entire section rewritten)

[revised manuscript text omitted]

**Commentaire [GY36]:** Comments 4

**Commentaire [GY37]:** Comment 6

**Commentaire [GY38]:** Comment 2

**Commentaire [GY39]:** Comment 5

**Commentaire [GY40]:** Comment 4

**Commentaire [GY41]:** Comment 1 from reviewer #1

a complementary tool to map these areas, as well as others for which no ice jam has yet been reported. And the model is a valuable tool for provinces or countries where no ice jam database exists.

For a future version of the model, potential developments could be:

- To consider attenuating factors, such as a section located immediately downstream a reservoir or directly within a rapid;
- To consider the width, shape and length of the contributing reach upstream from a predisposed section (is there potentially enough incoming ice to produce a jam?);
- To consider sudden channel widening (dissipation of the energy and ice run stalling);
- To take into account the presence of hydraulic structures (weirs, dams, dam reservoirs, etc.);
- To test the model using the US Ice Jam database (Carr et al, 2015)
- And certainly to use the slope index, upon availability of accurate elevation data.

The authors are presently starting the application of the model on all rivers prone to ice jams in the province of Quebec. They are also planning the work on a new version that will be integrated within an ice jam vigilance and alert system, combining spatial predisposition, temporal forecasting and ice status.

**Competing interests**

The authors declare that they have no conflict of interest.

**Acknowledgements**

The research presented here was funded by a NSERC Discovery Grant (2009-2014) to Dr. Bernier (INRS). The setup of the ice jam dataset was funded by the Quebec Ministry of Public Safety (MSP). The authors would also like to thank Jimmy Poulin and Fatou Sene from INRS and Nicolas Gignac from the Quebec Ministry of Public Safety for their contribution.

**References**

Banshchikova, L.S.: Monitoring of the Ice Jamming Process in Rivers Using Spatiotemporal Plots of the Water Levels, Russian Meteorology and Hydrology, 33(9), 600-604, 2008.

Beltaos, S.: "Chapter 6, Onset of breakup", River Ice Breakup (S. Beltaos, Ed.), (2009), Water Resources Publications, LLC, 480 pp.

Commentaire [GY42]: Comment 1

Commentaire [GY43]: Comment 1

Commentaire [GY44]: Comment 1 52

Commentaire [GY45]: Comment 7

Commentaire [GY46]: Comment 6

Commentaire [GY47]: Comment 3

[revised manuscript text omitted]

**Commentaire [GY54]:** Figure improved

 **Summary of comments from Reviewer #1 with initial reply from Authors**

*General comment*

*This paper explores ice jam predispositions along northern rivers using a geospatial modelling approach in which sets of fluvial geomorphological parameters are compared with ice jam occurrences. There is a high success rate of predicting ice jam locations, however some errors do occur due to the presence of sand bars and low water depths, variables not considered in the model The approach*
 *does give a first assessment of the ice jam potential of rivers, hence, the paper is deemed publishable if the following minor revisions are considered.*

Authors' response

We thank the reviewer for his comment. It is true that the simplified model presented here gives only a first assessment of the ice jam potential of rivers. Hence, the paper shows that even with limited data, it is possible to get a good sense of the areas at risk for ice
 jamming. This work can then be further used to build a version of the model that would better take into account some local and more complex causes of ice jams. This was better explained in the text.

Specific comments

*The narrowing index (NI) for bridge peers is rather arbitrarily derived that can lead to over- or under-estimation of their effect on ice jamming. No consideration was given to the number of peers spanning across the bridge. Hence, the NI of a suspended bridge would have*
 *the same NI value as a bridge with many closely spaced peers. Could you please give an explanation of why this wasn't considered?*

Authors' response

This aspect was of course considered in the development of the model. However, the information about the different characteristics of the bridges is not always available or easily accessible. Therefore, to maintain the objective of a simplified model that can be quickly deployed on many rivers, we have decided in this version, to consider all bridges equal. On a local scale, one could easily take a bridge out of the
 analysis if he considers that the structure is not a factor of ice jamming. An improved version of the model would certainly have to take bridges characteristics into account. This was better explained in the text.

Specific comments

*Also on the subject of bridges, I find that bridge peers do not necessarily stop an ice run to create an ice jam but reduce the inertia of the ice run enough for it to slow down and stop at a location further downstream from a bridge peer. Would such a consideration improve the*
 *predictability of the model?*

Authors' response

In its present version, the model considers a bridge to be an "aggravating factor" (coming either from obstacle or constraint). It uses "narrowing" as a strategy to apply this aggravating factor on a geospatial point of view.

**Summary of comments from Reviewer #2 with initial reply from Authors**

520 *Comment 1: "This paper presents a very interesting and original model that uses qualitative, geospatial information to quantitatively identify channel locations where ice jams could form. Its development and calibration is supported by an ice jam database. Overall, it seems that there is a great potential for the development of a model that could be used to identify potential ice jamming sites and this could be combined with a river ice breakup forecasting model. However, the model overlooks or simplify a number of key ice jamming parameters, factors, processes, and information that may limit its reliability and the paper really presents what appears to be the early*

525 *development stage of an acceptable model. "*

Reply 1: We really thank Dr Turcotte for the thorough review of our paper. Here we address every comment, suggestion or correction.

*Comment 2: "It seems that the authors globally lack in experience and confidence (a lot of sentences seem defensive) and the following points (below) should be considered to improve the next versions of the model (the actual knowledge about river ice processes has only be partially considered). I consider that the actual version of the model is almost dangerous to use by public security services or for flood*

530 *insurance purposes. "*

Reply 2: The authors of the paper are specialized in geomatics and remote sensing and have worked for many years on developing tools to support river ice and ice jam monitoring and characterization. However, they do not pretend to be experts in ice jam processes. This may explain the cautious (rather than defensive) tone of the paper. As for its "potentially dangerous" nature, we had better explained the limitations of the model in order to avoid any misinterpretation of the results.

535 *Comment 3: "At this point, I am not sure if I recommend (1) that the authors should make multiple technical changes to the paper and include a discussion that mentions the many limitations of the model in its actual form or (2) that the authors should present a new paper with a more advanced version of the model that would address and include most of the following points. I would tend to vote 1 because I consider that the model is original and represent a step forward in the field of river ice and flood forecasting. In this case, I would encourage the authors to present an improved version of the paper with a serious discussion and to present, in the years to come, an*

540 *improved version of the model that would potentially include a completely new model structure."*

Reply 3: We understand the ambivalence of the reviewer and we appreciate the opportunity he offers. Indeed, we think that the approach presented in the paper is innovative and that it produces very promising results. Its publication at this stage could help improve and validate the model within the river ice and public safety community. Discussion and conclusions sections were improved.

*Comment 4: "1. The authors do not mention that they have observed ice jams and do not refer to any experience in the field (e.g., to verify*

545 *the sites presented at Figures 8 – 10 or to look under bridges if pillars can be pointed out for ice jamming) . Therefore, this research is only based on theory and the authors cannot really confirm that the model is reliable. "*

Reply 4: Here we need to clarify a point (and it was done in the paper as well). The proposed model is not a physical model simulating the processes of ice jamming. Yes it is theoretical, in the sense that it is based on some common knowledge expressed by experts in the literature, about the general causes of ice jams. The model tries to express these causes in terms of channel morphology, within a 2D

550 spatial representation. Being develop for a wide application, the model uses simplifications and provide what we could name "first level" results. Although it could certainly be helpful, we do not think that going in the field was essential to this work at this point. But the model could certainly be fine-tuned for a specific river, with high resolution data and knowledge of local phenomena. Finally, the validation of the model is based on real events, not theoretical events, even if the historical database may contain some uncertainties. Therefore, we can certainly assess the model's reliability.

555 *Comment 5: "Beyond the government ice jam data base, the authors should have conducted a complete historical research and confirmed that the mentioned ice jam dates corresponded to specific hydro-meteorological events. This type of data base often confuses ice jams with other ice processes that generate winter or spring flooding (e.g., anchor ice and hanging dams). Moreover, at locations where observation is not easy, where there is no societal vulnerability, or where the jamming and release occur at night year after year, ice jams may have gone unnoticed (as somewhat mentioned in the paper). "*

560 Reply 5: We agree that the historical database is not perfect. We will better explain its limitations right from the start, in section 3.1.4. But it is nonetheless a unique source of information. The first author of the paper has worked with the government on the transfer and integration of historical observations within the database for the 3 rivers under study.

*Comment 6: "2. The authors refer to particular factors influencing ice jamming, but do not seem to understand all the physics that link these factors with ice jam processes. The authors never refer to the distinction between the toe and head of an ice jam and they barely*
565 *mention something about their potential length (that can be much greater than 250 m and therefore extend in sections that have nothing to do with the initiation of the jam). It is crucial to point out that the parameters influencing ice jamming sites refer to the toe (initiation site), which can be hundreds of meters or even kilometers away from the ice jam observation site. This can influence the results of the research positively or negatively. "*

Reply 6: We agree that we put the emphasis of the paper more on the geospatialization aspect than on the explanation of the ice jam
570 physical processes. We have given some justification for this in Reply 4. Background section improved.

*Comment 7: "3. A number of parameters such as channel widening (dissipation of the energy and ice run stalling), the presence of hydraulic structures (weirs, dams, dam reservoirs, etc.), and the presence of a tight, single bend (not a meander) have not been mentioned in the study and could help reducing false-negative errors."*

Reply 7: From the literature, these were not found to be part of the major factors causing ice jams. Added as potential developments.

575 *Comment 8: "On the other end, it seems that channel narrowing is assumed to generate ice jamming but in some cases, the concentration of energy actually favors the transit of an ice run."*

Reply 8: Again, the narrowing of the channel is mentioned in all references, as a major factor favoring ice jams. Thus the importance it is given in the model.

*Comment 9: "From my point of view, trying to fit many parameters in a "narrowing equivalent" will limit the potential development of the*
580 *model."*

Reply 9: Even if we fit many parameters into a unique narrowing index, each parameter is calculated independently and its relative importance can be adjusted. Also, representing each parameter as a narrowing equivalent simplifies the geospatial calculations. The final Narrowing Index is not just about a physical narrowing of the channel. It can also be viewed as a way to take into account, different aggravating factors.

585 *Comment 10: "4. Obstacles and gradient variations could explain a significant ratio of ice jams. This may require a more sophisticated spatial analysis that may become tedious to automatize. "*

Reply 10: Introduction and background improved.

*Comment 11: "5. One important parameter affecting ice jamming is the potential quantity of ice, i.e., the contributing reach. If there is not enough ice to produce an ice jam that can affect the floodplain, the jam may remain unnoticed. The most critical jamming sites are located*
590 *downstream of long sections where an ice run would simply not stop. This has to be mentioned here and potentially included in a future version of the model."*

Reply 11: We agree. Added in the discussion. Technically, long sections could be spatially identified.

Comment 12: "6. The model could consider factors that prevent the formation of an ice jam (e.g., immediately downstream of a reservoir) and the model could gain in accuracy and reliability. "

595 Reply 12: In a future version, we can certainly consider some attenuating factors for some river sections, not just aggravating factors. Added in discussion/conclusion.

*Comment 13: "7. In the end, at this development stage of the model, for Quebec, the data base itself could represent a more reliable tool to identify potential ice jamming sites than the model calibrated with the data base."*

Reply 13: This may be partly true because in any domain, if you have all the data, you do not necessarily need a model to replace the data.
600 But here, we know that the model will identify sections potentially predisposed to ice jams, even if no ice jam has been observed or reported yet. Also, the historical database does not cover all rivers in the province of Quebec. And such a database is not yet available in other provinces or in many countries. Hence, this shows the usefulness of the model, on top of the database. Added in discussion/conclusion.

*Comment 14: "There is no introducing context in the abstract."*

605 Reply 14: Added.

**Specific comments**

*Comment 15: "Line 7: "any" should be moderated. The model has been tested on three rivers only and a number of parameters and factors are not considered."*

Reply 15: This is the goal (to develop a model that can be applied on any river), not necessarily the end.   Changed.

610   *Comment 16: "Lines 16-17: I am not sure that talking about "false positives" is pertinent here. These are not really errors. "*

It is true, as we mention in the paper, that "false positives" are not really errors. We could change the term to "potentially false alarm" or "false positive cases".

*Comment 17: "Figure 1: Second part of the Figure may not be necessary. There is a lot of empty space in the map north of Quebec City that could be used to increase the size of the legend or to rearrange the ratio of each sub-Figure. Please confirm that the L'Assomption*
615   *River watershed is the right one. It seems that it is in contact with the St. Lawrence along 80 km. The southern part of the St-Francois River could be indicated approximately."*

Reply 17: Figure 1 improved.

*Comment 18: "Background: This section should be reorganized: The authors should mention that an ice jam can form because of congestion or because of an obstacle. The use of a transport capacity in the literature only represents one simplified interpretation that*
620   *has been overused here. Most of this section refers to congestion processes as if an ice jam could only be the result of an unimpeded ice run (Jasek 2003) that slows down and stop. Indeed, an important portion of ice runs encounter a physical obstacle (such as an intact ice cover mentioned at the end) and suddenly stop. This has nothing to do with congestion or a "reduction in the ice transport capacity". "*

Reply 18: As mentioned in reply 10, background section modified accordingly.

*Comment 19: "Line 39: I am not sure if Beltaos would refer to "volume of ice". I believe that it would be the "ice discharge". "*
625   Reply 19: No. the term used by Beltaos is "ice volume".

*Comment 20: "Line 41: Authors should seek additional references. The books "River ice Jams" or "River ice Breakup" are potential sources of complementary information. "*

Reply 20: We have used these two books. More references were added.

*Comment 21: "Lines 43-44: The authors should mention that there are different types of islands. They can be naked bars, vegetated bars*
630   *or emerging rock outcrops (vegetated or not) and not all of them are associated with a break in the channel gradient. From my point of view, the presence of an island is associated with ice jam in part because the flow can bypass the congested channel and release the pressure on the impeded ice run, therefore leaving an ice jam on one side of the island. The authors only mention "narrowing" as the basic process to identify potential ice jamming sites. "*

Reply 21: We didn't want to infer that the only ice jam related physical process involved in the presence of an island was narrowing. But
635   "narrowing" is how we "conceptualize" the impact of islands in our simplified model. Clarifications were made.

*Comment 22: "Lines 45-47: Not all bridges present pillars and pillars are often profiled to minimize to effect on flow conditions and ice transport capacity. Also, bridges are often build at natural (or artificial) narrows that already represent a limitation for ice mobilisation.*

*This may affect the result of the study. Also, the flow often accelerates under a bridge because of the smaller river width and ice runs could easily transit that these locations. "*

640 Reply 22: This was acknowledged in reply to reviewer #1. Here the bridges are considered equal because for application of the model on a large area, the information about individual bridges is not always available. But at the local level, one could simply take individual bridges out of the analysis if he knows it is not an aggravating factor.

*Comment 23: "Lines 48-50: Not well explained. The basic process may be that the flow along the concave bank drowns while the ice floats. Also, the authors mentioned that sinuosity may "initiate" a jam. This is correct, but it would mean that the first bend of a*
645 *meandering reach is more likely to cause jamming that the last one. The authors could include this (or mention that this has been or should be considered) in their study. "*

Reply 23: Clarification was made.

*Comment 24: "Lines 57-59: This is one of the most important parameters explaining ice jamming. "*

Reply 24: Agree. But as we will see later, it requires accurate elevation data.

650 *Comment 25: "Lines 60-62: There are many types or gravel bars in gravel bed rivers. The authors only mention two types here (point bars or side bars). I am not sure that I agree with the reasoning presented: gravel bars are usually mobilized when the flow increases and stabilize when the flow decreases. How could they form and migrate if there was never a potential for transport? The first part of these lines does not need a reference as most people know about bars. It is really the second part of the sentence that needs the support of a reference."*

655 Reply 25: Clarification was made.

*Comment 26: "Lines 65-66: The authors should refer to the concept of an impeded ice run here (Jasek ,2003, or Jasek and Beltaos, 2008). This is the only parameter that does not directly refer to the morphology, but it is very important. This is why I would reorganize this entire section. "*

Reply 26: The entire section was strengthened. However, we should again keep in mind that we do not prepare the reader for the
660 development of a physical model. We only want to give him a basic understanding of how morphology can play a role in ice jam formation.

*Comment 27: "Line 69-70: This is important because the authors use this single idea of the combination of two ice jamming factors for their model. I understand the need to simplify reality, but I am not sure that one publication can justify this choice. "*

Reply 27: The idea here is that an ice jam formation is often due to a combination of different factors, as seen in the literature. The study
665 from Kalinin presents numbers that reinforce this idea. Thus, our model combines and weight different parameters (natural narrowing, convergence with a tributary, presence of a bridge, presence of an island, sinuosity). Clarification was made.

*Comment 28: "Line 77-78: I would express this differently. The word "dynamic" in the river ice literature usually refers to processes such as ice runs, ice jams and ice dams. Note that the depth can be linked to the morphology and the cover characteristics as well (e.g., Turcotte and Morse, 2013). If the authors believe that the presence of bars is important, well their emergence is completely linked to the depth and discharge!"*

Reply 28: Dynamic has been replaced by variable. As for bars, refer to Reply 25 and 26. Here, the comment from the reviewer is correct. Bars are dependent on the water level and discharge. Which are variable throughout the year and more so during the spring. This is why we do not consider the bars in this model.

*Comment 29: "Line 87: How does the 250 m in length compares with the channel width? Why not using a variable length that depends on the width of the channel or the homogeneity of the morphology and alignment? I guess that this would be complex for automatic interpretation to be performed and it would not fit with the title of the paper."*

Reply 29: The width of the channel was not a factor when deciding the length of the river sections used for the model. It had more to do with the resolution of the input data and the scale of the parameters we were calculating. For example, we could average the channel width every 5 meters if we wanted to. But we would get micro narrowing. And the sinuosity must also be estimated over a certain distance to be significant. On the opposite, an ice jam can run on several hundreds of meters or even several kilometers. This would be too coarse for the scale of the model. We decided to compromise with 250m sections.

*Comment 30: "Line 88: What can be said here about the size of ice jams if the data base does not include such information?"*

In the historical database, ice jam length is only mentioned on some occasions. But from observations and from literature we know that ice jam length can vary from hundreds of meters to kilometers.

*Comment 31: "Line 89-91: Is this precise enough to document parameters such as narrowing?"*

Yes. The planimetric accuracy of these dataset is better than 2m. Clarification was made.

*Comment 32: « Line 92: Islands: Does the model differentiate bars and stable islands?"*

Reply 32: Metadata from the data provider do not mention what was digitized as an island. But looking at the island vector over Google Earth clearly shows that for the three rivers of this study, islands correspond to "vegetated islands". Clarification was made.

*Comment 33: "Line 92: rapids: This is very important and has not been mentioned before. Ice jams almost never initiate in rapids but often at the end of rapids. Does this include riffles or just rapids?"*

Reply 33: Again, metadata do not inform about the types of rapid. But overlaying this layer to Google Earth seems to indicate that it concerns rapids, not small riffles that may disappear with higher water level. We will add a word on rapids in the improved background section, when discussing slope changes. It would also be possible to force a low predisposition to sections in a rapid. Clarification was made.

Comment 34: "Line 93: This is not very reassuring: The authors should mention that a width less than X m could not be included in the model for spatial information accuracy limitations. Then, it means that the model is actually not adaptable to small rivers. "

Reply 34: Metadata from the data provider do not mention the minimal channel width for which they use a polygone. We have checked the three rivers in this study and all sections in polygon format are at least 20m wide. The text was clarified accordingly.

*Comment 35: "Lines 100-102: I understand that the model is simplified for practical reasons. However, as noted in the general comments and as the authors mention at the end, these four factors are linked to ice jamming processes for distinct physical reasons. "*

Yes, the physical reasons are different. But the model only needs to know where it occurs (where is the natural narrowing, the bridge, the tributary, etc…) and what aggravating factor to apply at this place.

*Comment 36: "Line 104-105: About the secondary channel presenting a more competent ice cover: I do not agree and the authors do not refer to any study to support this. From my point of view, there could be less (or no) ice in the secondary channel and at some location, the secondary channel plays a determinant role in the ice jamming initiation process. "*

Reply 36: The assumption was that the main channel (here the more direct route) is the one with the maximum discharge. Hence, the secondary channels would freeze earlier. This section was modified.

*Comment 37: "Line 105-107: Every island site is different and I am not sure that the simplification proposed by the authors is the most adequate one. Food for thought."*

Reply 37: Initially, we tried to consider in our analysis, the position of the island in the channel as well as the shape of the island. This proved to be complex and hard to calibrate. This is why we went for such a simplification. Clarification was made.

*Comment 38: "Line 108: Do you have any information about the pillars? What is the assumption here? If the bridge is located downstream of rapids, ice blocs may be small and easily pass under. If large is slabs come in contact with wide, rectangular pillars, yes, they might be stopped right there. This would be a serious engineering error that as no real link with a channel narrowing. Line 109: What do you mean by "initially"? I believe that this factor should have been calibrated more accurately or considered differently (not a narrowing). Line 112-113: I believe that this is a major mistake made by the authors: Presenting an assumption in the methodology, mentioning that it could be improved before the results are presented and not doing anything later despite this could have been better calibrated."*

Reply 38: As mentioned before, there is no physical assumption here, other than that the presence of a bridge may increase the possibility of an ice jam (either as an obstacle or a constraint). Again, bridges that do not pose such a risk should be taken out of the analysis. The weight of the remaining bridges could be adjusted like for instance, one could adjust the manning coefficient when trying to better fit a hydraulic model to a specific river. On the Chaudière River, there are 18 sections with a bridge. Due to the weight applied to bridges, all sections are classified as having high (15) or medium (3) predisposition. Of these 18 sections, 7 report ice jams. So this indicates that the model is right to consider bridges as an aggravating factor but at the same time, that not all bridges are equal and that fine tuning should be done at the local scale. Clarification was made..

*Comment 39: "Line 116: Specify that this gives more importance to large tributaries. Again, this has almost nothing to do with channel narrowing."*

Reply 39. Again, it is true that the impact of the tributary is just indirectly related to the concept of narrowing. The strategy of considering it as a narrowing is a scheme to apply a predisposition weight at this spot.

*Comment 40`"Figure 3: Does not explain well how the tributary is considered"*

Reply 40: Figure improved.

*Comment 41: "Figure 4: The flow direction should be the same than in Figure 3. A narrowing index (equation) should be presented for each presented section."*

Reply 41: Figure improved.

*Comment 42: "Line 131-132: This definition refers to SV, not the Sinuosity4. . . The authors should mention that SV is always larger than 1."*

*Comment 43: "Line 133: I am not sure that "several" will satisfy the reader. Can you present a range? Can this take into account single bends and not only meanders?"*

Reply 42-43: This part was correctly rephrased.

*Comment 44: "Line 139-142: Did the authors try to use the 5 m resolution? I am sure that some governmental agencies have data concerning river profiles and hydraulic models. This would probably not be precise enough to determine changes between 250 m sections, but it could very well identify slope breaks that are so important in jamming processes. As a reader, I am disappointed about this ending and this introduces a difficulty acceptable omission in the model. "Luckily" for the authors, a change in slope is normally characterized by a change in morphology and pattern and therefore slope breaks are somewhat indirectly covered by the model. Including gradient data would probably improve the model's result."*

Reply 44: Yes, in the development phase, we used the 5m elevation data to detect slope breaks. A DEM was created by interpolating contour lines. Then, altitude was extracted at the upstream and downstream limit of each 250m section to calculate slope. However, due to accuracy of the data and to the interpolation process, this would result in a longitudinal profile showing big steps. At the time, river ice experts advised us to take out the slope index. If lidar data were to become available over an entire river, it should be possible to reintegrate the slope index. The link between slope and channel morphology may be a reason why even without a slope index, we can obtain very promising results. Clarification was made.

*Comment 45: "Line 146: I would say "approximate" since most jams are longer than a single coordinate and because this geolocation does not refer to the toe where the jamming process is initiated. In some instances, the toe could be kilometers away from the observation point. Also, as mentioned in the general comments, some reported ice jams could be intense anchor ice or frazil jam events and these*

*processes could take place at locations where ice jams are not likely to form. A validation with a corresponding rising Q should be performed.*

Reply 45: Clarification done. Concerning the possible anchor ice or frazil jams, it is out of the scope of this study to validate the historical database.

*Comment 46: "Line 157: This is the Chaudière River section but this sentence refers to the three rivers."*

Reply 46: The model was developed mainly with the data from the Chaudière River. However, to determine the thresholds for the classes of Narrowing and Sinuosity index using K-means, we have decided to use the entire range of values from the three rivers in this study. This provides a more robust and representative model. Clarification was made.

*Comment 47: Lines 163-166: Note here that you use "reported" ice jams to calibrate a model. Taking into account previous comments may improve the calibration result and their reliability.*

Reply 47: Clarification was made.

*Comment 48: "Table 3: This is a fair analysis tool but note that this version of the model cannot gain a high precision potential in part because it is limited by a 2D IJPI. Also, note that highly sinuous reaches often present a relatively constant width and therefore, the two parameters considered here are not independent. Therefore, a value of 1 may be difficult to obtain in reality."*

Reply 48: We do not understand this comment and to what it refers.

*Comment 49: "Line 201: About false negative errors: Please consider the potential length of the jam and the difference between the initiation site (predisposition) and the observation site (anywhere along the jam). Line 202: About false positive errors: Not really an error because this is not an ice jam temporal prevision model. Please consider at everywhere in the paper that important ice jams can happen where there is no observation point nor vulnerability.."*

Reply 49: We agree. Clarification was made..

*Comment 50: "Line 209: I understand that the model can underestimate some factors, but this is your calibration river. This could have been better considered if you were not limited to two indicators and if observations had been made in the field."*

Reply 50: The model has two indices but they in fact reflect five factors that can favor the initiation of an ice jam. Tributaries being one of them.

*Comment 51: "Lines 210-211: Exactly, and this should be stated clearly in the previous sections. Not only a source of incoming ice, but also a source of incoming runoff and javes."*

Reply 51: Clarification was made.

*Comment 52: "Lines 221-222: Please eventually consider: The contributing area for ice blocks, the gradient, an increasing width, the absence of observation points along the river, the absence of vulnerability along the river, and finally, ice scars on trees. You should present the potential reasons for false-positive errors in the form of bullets."*

Reply 52:  Added to the discussion.

*Comment 53: "Line 219: Same comment as line 209. Bridges are often located at natural narrows, but there design does not necessarily impede the transit of ice runs."*

Reply 53: This point was addressed earlier.

*Comment 53b: "Line 224: How short? Please specify a range."*

 Reply 53b:  Clarification done.

*Comment 54: "Table 4: You could have further investigated the 6 "no specific feature". This could mean factors that have not been considered.*

*"Table 5: Same comment as Table 4"*

*"Lines 259-263: The model may be missing something because there is no info about the gradient. A widening is also a site for ice jams, especially downstream of rapids or riffles. Note also that the analysis assumes that the toe of the jam was correctly located, which may not be true."*

Reply 54: This was better stressed out in the discussion.

*Comment 55: "Line 245: Note about bridges: Their presence can mean less snow ice and more thermal ice. Their presence can also be associated with de-icing salt falling on the ice surface. Then, what would be the final relative ice resistance at the time of breakup?"*

Reply 55: This is a very local consideration and cannot be accounted for in a general model.

*Comment 56: "Line 260: Please add a reference that supports that sand bars are associated with ice jams. I do not know any."*

Reply 56: Refer to reply 25.

*Comment 57: "Line 265-266: Yes. But then, this could be associated with other morphological or areal patterns."*

 Reply 57: True but still a time dependent parameter.

*Comment 58: "Lines 278-280: A lot of sections may be associated with ice jamming, but the contributing area is just too small. The ice jam will most often form in the first (upstream) predisposition area located downstream of a long stretch of relatively fragile ice."*

Reply 58: Interesting point. Could be considered in a future version of the model.

*Comment 59:*

810 *"Figure 8: Potential interpretation (I do not know this site): This is enough narrowing, widening and changing direction to generate an ice jam. The low floodplain on the left bank and the possible secondary channel all support ice jamming. The marker may point a pool between two riffles where ice jams often form (against a small hanging dam. . .).*

*Figure 9: Potential interpretation: B: The ice run loses energy in the bend and loses further energy in the widening. C: Water evacuation channels and changes in direction, enough to initiate an ice jam.*

815 *Figure 10: Potential interpretation: This is an energy concentration area followed by a dissipation area. If there is no info about the jamming scenario (jamming of an impeded or unimpeded ice run), the reason for the ice jam event is hard to certify."*

Reply 59: This was considered in the new conclusion section.

Comment 60: *"Figures 8 to 10: Note how all these reported jam are located where roads or houses are close to the river, ideal observation points that may not correspond to the ice jam toe."*

820 Reply 60: See reply 47.

Comment 61: *"Table 6: You could lower the False-Positives by considering the length of the potential contributing area."*

Reply 61: See reply 58.

Comment 62: *"Line 300 and Lines 302-303: Yes, but the slope could not be considered and this should be mentioned as a potential development, not as a limitation of the actual model. This should be part of a discussion section. Overall, some sentences could be written 825 more positively and confidently."*

Reply 62: Clarification done.

Comment 63: "Line 314: Here, the bathymetry is considered as a dynamic parameter. Why? Why not just considering morphological (pattern, width, gradient, topography) and ice (type, potential thickness, possible processes) parameters?"

Reply 63: By bathymetry, we mean water depth. As with ice, it is a variable parameter that cannot be considered in this simplified model.

830

---

## Author Response (AR2)

**River predisposition to ice jams: a simplified geospatial model**

Stéphane De Munck, Yves Gauthier, Monique Bernier, Karem Chokmani, and Serge Légaré

**Iteration: Minor Revision**

All requested minor corrections have been done.

Line 9: The word "breakup" is usually accepted and should be consistent in the paper whereas "freezeup" or "freeze-up" are both acceptable. **Done**
Line 9: predicting "the timing of" river ice breakup … **Done**
Line 16: Results show "that" 77% … **Done**
Line 32: Remove "some" **Done**
Figure 1: This Figure is much better than the previous one. Please adjust the orientation of the North arrow
Line 36: , "all tributaries of the St. Lawrence" River. **Done**
Line 45: in resisting forces to ice transport, including impeded ice runs pushing against an intact ice cover. **Done**
Line 46: use "resisting" **Done**
Line 46: "directly and indirectly" governed by **Done**
Line 50: Excellent
Line 52: "and will likely present a thicker, more resistant ice cover at breakup" **Done**
Line 54: remove "bottom" **Done**
Line 55: which represents an additional resistance to lifting and mobilization when the discharge increases. **Done**
Paragraph 43-65: Consider splitting this into two or more smaller paragraphs. **Done**
Remove line 72-73 as this is included in your list and explained earlier. **Done**
Move paragraph Line 74-77 above your selected or summary list. **Done**
Line 78: "will" should be "should" **Done**
Line 83: "will" should be "are" **Done**
Line 83: Last sentence should be: "This is a reasonable assumption since the presence of a thick ice cover can be linked to morphological indicators, as proposed by river ice conceptual models (e.g., Turcotte and Morse, 2013)." **Done**
Line 96: "but it would be…" should be "and could be implemented in a subsequent version of the model" **Done**
Lines 96 to 107: You mention the number of sections at the beginning and then mention the spatial limitation to finally present the length for each river. From my point of view, this could be better organized. **Done**

Lines 120-121: "release of pressure when the ice run and some water is deflected into a secondary channel" **Done**

Line 128: Remove the sentence that starts with "And" **Done**

Line 130: "when a bridge is crossing…" should be "at bridges would give them an adequate weight in the final…" **Done**

Line 135: "run off" should be "runoff" **Done**

Line 138: "the ice run can stop at the confluence to form an ice jam that could subsequently intercept subsequent ice runs from the main channel to form a larger ice jam" **Done**

Line 155: "does not" **Done**

Line 156: "which could be the case in reality." **Done**

Lines 163-164: use "is" instead of "was" to be consistent with the preceding sentence. **Done**

Line 174: Merge the two sentences **Done**

Line 178: "at the end" should be "at their foot" **Done**

Line 178: "force" could be "manually impose" to sections with "known" rapids **Done**

Section 3.2.1: Comment: I don't see a problem regarding using as much information as possible about confirmed ice jam locations in order to calibrate the model independently for any river. This would mean that the weight would be river, reach or morphology specific. I would make it more robust and reliable. The authors should consider this when applying the model to multiple rivers. **ok**

Line 254: I believe that "so" could be replaced by a more appropriate "therefore", "as a result", "in this case" etc. Also, throughout this section, there is some redundancy and this (lines 247-255) could be more efficiently expressed. **Done**

Line 256: "Table 4 finally shows that 32 sections (7%) where classified with a high predisposition…" **Done**

Line 258: Merge the two sentences **Done**

Line 264: remove "the" **Done**

Line 277: "a closer look at some false-negative errors that are important in terms of public safety because of adjacent vulnerability." **Rephrased**

Line 280: From my point of view, these are more "bars" than "islands". **Done**

Line 292: "not directly considered by the model" **Done**

Line 292: Two sentences starting with "again". This should be merged. **Done**

Line 318: merge two sentences with "since" **Done**

Line 332: Merge two sentences with "and" **Done**

Line 336: would become available **Done**

Line 354: "new version of the model" **Done**

The reviewer also suggested, in very general terms, that the paper should be shared with colleagues in order to improve the efficiency of some sections and maximize the impact of the paper. We think that the specific submission/public reviewing process of NHESS is sufficient to achieve this goal.